# Conserved and divergent development of brainstem vestibular and auditory nuclei

Marcela Lipovsek[1]*, Richard JT Wingate[1,2]

[1]Centre for Developmental Neurobiology, Institute of Psychiatry, Psychology and Neuroscience, King's College London, London, United Kingdom; [2]MRC Centre for Neurodevelopmental Disorders, Institute of Psychiatry, Psychology and Neuroscience, King's College London, London, United Kingdom

**Abstract** Vestibular function was established early in vertebrates and has remained, for the most part, unchanged. In contrast, each group of tetrapods underwent independent evolutionary processes to solve the problem of hearing on land, resulting in a remarkable mixture of conserved, divergent and convergent features that define extant auditory systems. The vestibuloacoustic nuclei of the hindbrain develop from a highly conserved ground plan and provide an ideal framework on which to address the participation of developmental processes to the evolution of neuronal circuits. We employed an electroporation strategy to unravel the contribution of two dorsoventral and four axial lineages to the development of the chick hindbrain vestibular and auditory nuclei. We compare the chick developmental map with recently established genetic fate-maps of the developing mouse hindbrain. Overall, we find considerable conservation of developmental origin for the vestibular nuclei. In contrast, a comparative analysis of the developmental origin of hindbrain auditory structures echoes the complex evolutionary history of the auditory system. In particular, we find that the developmental origin of the chick auditory interaural time difference circuit supports its emergence from an ancient vestibular network, unrelated to the analogous mammalian counterpart.
DOI: https://doi.org/10.7554/eLife.40232.001

*For correspondence:
marcela.lipovsek@kcl.ac.uk

**Competing interests:** The authors declare that no competing interests exist.

## Introduction

The colonisation of land by tetrapods led to a series of independent solutions to the problem of adapting sensory systems from water to air. In particular, the auditory apparatus was modified several times through a mixture of alteration of ancestral structures and *de novo* innovations (*Chagnaud et al., 2017*; *Fritzsch and Elliott, 2017*; *Fritzsch and Straka, 2014*; *Grothe and Pecka, 2014*; *Manley, 2000*; *Manley, 2017*). These changes also required the evolution of associated neural processing networks in the hindbrain. Each extant neural circuit is therefore an elaboration of the remarkably conserved brainstem groundplan (*Nieuwenhuys, 2011*) and has been resourced from a repertoire of neural lineages that depicts an ancestral rhombomeric arrangement (*Philippidou and Dasen, 2013*; *Wullimann et al., 2011*).

The interplay between ancestral and *de novo* responses to changing environmental context is readily apparent when comparing the auditory and vestibular systems, which have interrelated developmental and evolutionary histories (*Duncan and Fritzsch, 2012*; *Fritzsch and Straka, 2014*; *Manley et al., 2004*). Both systems process mechanical stimuli and share an ancestral receptor cell type (the hair cells), which sit in specialised sensory epithelia located in the inner ear and develop from the same ectodermal thickening, the otic placode (*Fritzsch and Elliott, 2017*; *Fritzsch and Straka, 2014*; *Whitfield, 2015*). Vestibular sensory input has remained mostly unchanged and, correspondingly, vestibular peripheral organs, hindbrain vestibular nuclei and their projection patterns are highly conserved across vertebrates (*Straka and Baker, 2013*; *Straka et al., 2014*). All

vertebrates possess a conserved set of vestibular sensory epithelia that project, via the eighth nerve, to a conserved set of hindbrain vestibular structures: superior, lateral (or Deiters), medial and descending (inferior or spinal) vestibular nuclei (*Figure 1A*).

In contrast, the transition from an aquatic to a land environment brought about a marked transformation of the auditory scene and led to various specialisations for the detection of airborne sound (*Carr and Christensen-Dalsgaard, 2016*; *Carr and Soares, 2002*; *Clack, 2015*; *Fritzsch and Straka, 2014*; *Grothe et al., 2004*; *Grothe and Pecka, 2014*; *Manley, 2000*; *Manley, 2017*). Paleontological, morphological, functional and behavioural evidence suggests that a number of auditory peripheral and central innovations emerged separately in the different clades of land vertebrates (*Grothe and Pecka, 2014*; *Manley, 2000*; *Manley, 2017*; *Manley et al., 2004*). Most striking amongst them is the independent emergence during the Triassic period, more than 100 million years after the separation of the tetrapod lineages, of at least five variants of a tympanic middle ear, which operates as an impedance matching device for the efficient detection of airborne sounds (*Anthwal et al., 2013*; *Carr and Christensen-Dalsgaard, 2016*; *Clack, 2015*; *Kitazawa et al., 2015*; *Manley, 2000*; *Tucker, 2017*). This was accompanied by the independent elongation of the auditory sensory epithelia, a parallel diversification of hair cell types and concomitant elaborations of hair cell based sound amplification mechanisms, ultimately leading to fine tuning of sound detection and expansions of the hearing range to higher frequencies in several amniote clades (*Dallos, 2008*; *Hudspeth, 2008*; *Köppl, 2011*; *Manley, 2000*; *Manley, 2017*).

Alongside the independent emergence of middle and inner ear innovations, amniotes have developed neural mechanisms for processing sound stimulus of increasing frequency with increasing accuracy. As a result, the complement of brainstem auditory nuclei, the central targets of the auditory

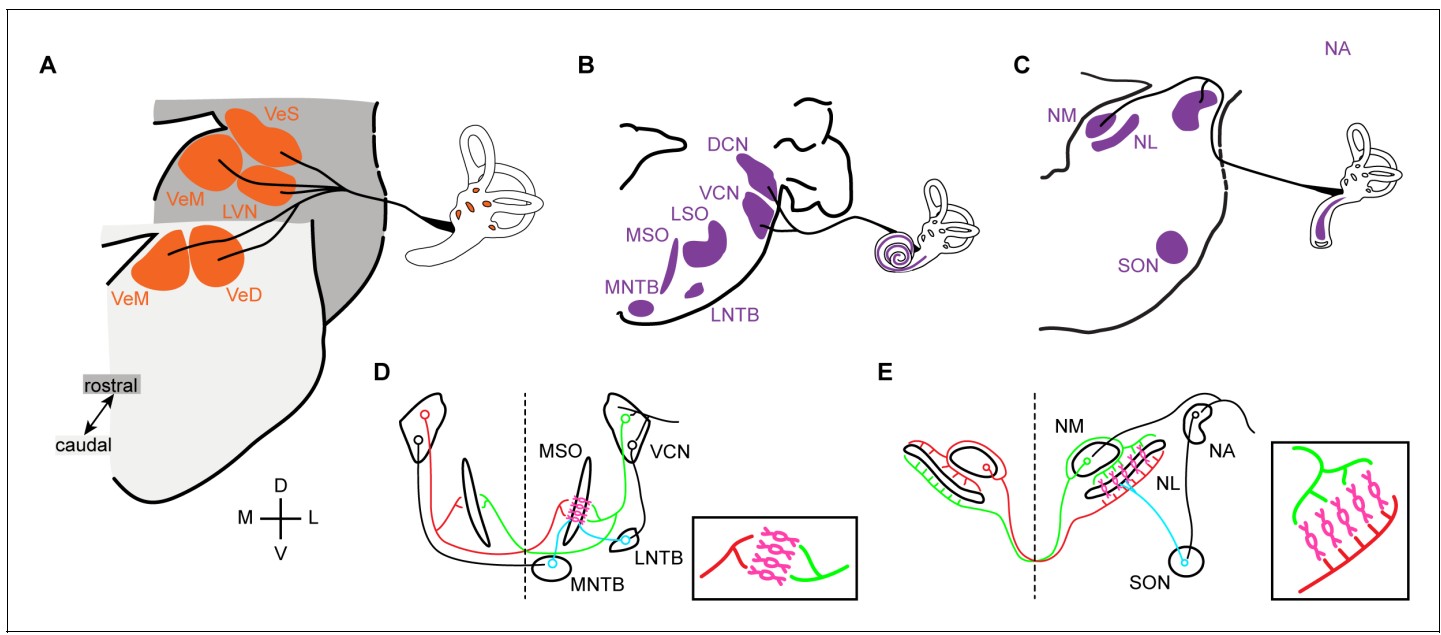

**Figure 1.** Vestibular and auditory brainstem nuclei of mammals and birds. (**A**) Schematic diagrams of hindbrain coronal sections showing the four main vestibular nuclei of vertebrates that receive direct input form the VIIIth (vestibular) nerve. VeS, Superior Vestibular Nucleus; LVN, Lateral Vestibular Nucleus (Deiter's Nucleus); VeM, Medial Vestibular Nucleus; VeD, Descending Vestibular Nucleus (Spinal or Inferior Vestibular Nucleus). D, dorsal; V, ventral; M, medial; L, lateral. (**B-C**) Schematic diagrams of hindbrain coronal sections depicting the mammalian (**B**) and avian (**C**) hindbrain first order auditory nuclei that receive direct input from the VIIIth (auditory) nerve and the main second order nuclei to which they project. DCN, Dorsal Cochlear Nucleus; VCN, Ventral Cochlear Nucleus; LSO, Lateral Superior Olive; MSO, Medial Superior Olive; MNTB, Medial Nucleus of the Trapezoid Body; LNTB, Lateral Nucleus of the Trapezoid Body; NA, Nucleus Angularis; NM, Nucleus Magnocellularis; NL, Nucleus Laminaris; SON, Superior Olivary Nucleus. (**D-E**) Schematic representation of the main connections of the mammalian (**D**) and avian (**E**) ITD circuits. *Insets.* Close-up view of the coincidence detection neurons showing the arrangement of ipsilateral and contralateral connections. Bilateral projection neurons, red and green. Coincidence detection neurons, magenta. Inhibitory neurons, cyan. Auditory nerve inputs and inhibitory connections are drawn only on the right side for simplicity.

DOI: https://doi.org/10.7554/eLife.40232.002

branch of the eighth nerve, varies greatly between clades and no true homologies could be established between them thus far (*Carr and Christensen-Dalsgaard, 2016*; *Grothe et al., 2004*; *Nothwang, 2016*). Mammals (the first clade to split from the ancestral amniote branch) possess two first order auditory nuclei, the ventral and dorsal cochlear nuclei. These project to second order hindbrain nuclei: lateral and medial superior olive and lateral and medial nucleus of the trapezoid body, as well as to other brainstem and higher order targets (*Carr and Soares, 2002*; *Grothe et al., 2004*) (*Figure 1B*). In diapsids (reptiles and birds), two first order nuclei receive auditory nerve afferents: the nucleus angularis and nucleus magnocellularis. These project as well to second order hindbrain nuclei (superior olivary nucleus and nucleus laminaris) and higher order structures (*Carr and Soares, 2002*; *Grothe et al., 2004*) (*Figure 1C*).

Notwithstanding the presence of such divergent arrays of hindbrain auditory structures, both mammals and archosaurs (birds and crocodilians) have developed mechanisms for sound source localisation based on hindbrain neural processing circuits. Functional analogies and convergent evolution have been proposed for some sub-circuits (*Carr and Christensen-Dalsgaard, 2016*; *Carr and Soares, 2002*; *Grothe et al., 2010*). An example is the interaural time difference (ITD) circuit that compares the difference in time of arrival of sound to each ear to determine the location of the sound source in the horizontal plane. In both mammals and archosaurs the overall organisation of the circuit is similar, with bilaterally projecting first order neurons that converge in coincidence detection neurons that are, in turn, modulated by inhibitory input (*Figure 1D,E*). However, a number of circuit features reveal that the functional analogy is only superficial and indicate that the ITD circuits are an example of convergent evolution (*Carr and Christensen-Dalsgaard, 2016*; *Carr and Soares, 2002*; *Grothe et al., 2004*; *Grothe and Pecka, 2014*; *Grothe et al., 2010*). Although coincidence detection neurons are present in both groups, only the avian connectivity resembles a Jeffres model circuit (*Jeffress, 1948*) (*Figure 1E*), with axonal delay lines that contact coincidence detection neurons arranged as a place code (*Ashida and Carr, 2011*; *Grothe et al., 2010*). Such a wiring pattern is not present in mammals (*Grothe et al., 2010*; *Karino et al., 2011*) (*Figure 1D*), and the ITD coding strategies differ between the two clades (*Grothe and Pecka, 2014*; *Grothe et al., 2010*). Moreover, inhibitory modulation plays different roles in avian and mammalian ITD circuits (*Grothe, 2003*; *Grothe and Pecka, 2014*).

The notable disparity between the highly conserved vestibular hindbrain and the independently elaborated auditory hindbrain provides a framework on which to interrogate the contribution of developmental processes to the evolution of neuronal circuits, given that both systems are derived from the ancestral octavolateral column. In particular, to what extent are the respective ITD circuits de novo innovations for processing sound localisation, or a convergent assemblage based on a similar repertoire of ancestral neurons?

To answer these questions, we perform a comparative analysis of the developmental origin of vestibular and hindbrain nuclei. We have exploited recent advances in genetic fate-mapping in chick embryos (*Green and Wingate, 2014*; *Kohl et al., 2012*) to compare the origins of avian vestibular and auditory neurons with that recently characterised in mouse (*Di Bonito et al., 2013*; *Di Bonito et al., 2017*; *Farago et al., 2006*; *Fujiyama et al., 2009*; *Marrs and Spirou, 2012*; *Pasqualetti et al., 2007*; *Yamada et al., 2007*). Specifically we have studied the rhombic lip lineage, characterised by a progenitor pool expressing *Atoh1* (*Ben-Arie et al., 2000*; *Machold and Fishell, 2005*; *Rose et al., 2009*; *Wang et al., 2005*), and a ventricular zone lineage whose precursors express *Ptf1a* (*Meredith et al., 2009*; *Yamada et al., 2007*). These two lineages collectively define the majority of the auditory circuit in mouse, including all first order auditory neurons (*Fujiyama et al., 2009*). To underpin the *Atoh1* and *Ptf1a* lineage contributions to avian hindbrain vestibular and auditory nuclei we performed an electroporation-based fate-mapping in chick embryos, using enhancer driven reporter constructs. Additionally, we traced their axial origin through the electroporation of reporter constructs driven by enhancer elements from *Egr2* (*Krox20*), *Hoxb1*, *Hoxa3* and *Hoxd4*.

Our results lend strong support to the hypothesis that vestibular nuclei are developmentally homologous in chick and mouse to a fine level of detail, reflecting the ancestral origin and conservation of these hindbrain structures. By contrast, auditory nuclei show a mixture of conserved and divergent developmental origin. In particular, functionally analogous circuit components in the ITD circuit of the chick and mouse have different lineage contributions supporting a long-held hypothesis that they represent an example of evolutionary convergence.

# Results

In order to determine the developmental origin of hindbrain auditory and vestibular nuclei, we performed *in ovo* electroporation of chick embryos at stages 12 to 14 (*Figure 2A*) with two contrasting sets of conditional reporter constructs. First, we used enhancer elements from the bHLH transcription factors *Atoh1* (*Helms et al., 2000*; *Kohl et al., 2012*) and *Ptf1a* (*Meredith et al., 2009*), which are expressed in distinct dorsoventrally located neuronal progenitors (*Figure 2B*). Electroporation with Atoh1-Cre + Flox-pA-GFP + CAGGS-mCherry (*Figure 2C*) or Ptf1a-Cre + PBase + Pb-Flox-pA-GFP + CAGGS-mCherry (*Figure 2D*) resulted in broad expression of mCherry and specific GFP labelling of cells located at the *Atoh1*$^+$ rhombic lip or more ventrally located stripes of *Ptf1a*$^+$ progenitors, respectively. The expression of *Atoh1* and *Ptf1a* revealed by *in situ* hybridisation (*Figure 2B*) was faithfully recapitulated by the enhancer-driven GFP expression from the reporter constructs (*Figure 2C,D*). Second, we used enhancer elements from four transcription factors whose expression maps onto segmental boundaries of the early, rhombomeric hindbrain: *Egr2* (*Krox20*), *Hoxb1*, *Hoxa3* and *Hoxd4*. The expression pattern of the segmental markers was recapitulated by the reporter constructs employed. An *Egr2* enhancer element directed GFP expression to r3 and r5 (*Chomette et al., 2006*); a *Hoxb1* enhancer element directed GFP expression to r4 (*Ferretti et al., 2005*); a *Hoxa3* enhancer element directed GFP expression to r5 and r6 (*Manzanares et al., 2001*) and finally a *Hoxd4* enhancer element directed GFP expression caudal to the r6/r7 boundary (*Morrison et al., 1997*) (*Figure 2E*).

To achieve a comprehensive anatomical identification of hindbrain structures, electroporated embryos were incubated to embryonic day 10 (E10) and the distribution of cells within hindbrain nuclei assessed in coronal cryosections. All structures were identified based on the fluorescent signal from the electroporated constructs, nuclear counterstaining and in reference to Nissl stained sections of equivalent orientation and stage (*Figure 3A*). Fluorescent label was contrasted to Nissl derived templates (*Figure 3B*) to identify nuclear boundaries on the basis of cytoarchitecture.

## *Ptf1a* and *Atoh1* lineages contribute distinct neuronal populations to hindbrain vestibular nuclei

The vestibular nuclear complex extends rostro-caudally along the entire dorsal hindbrain. Six vestibular nuclei are commonly recognised in birds: superior, Deiters dorsal, Deiters ventral, tangential, medial and descending (*Wold, 1976*). Vestibular neuronal groups are also defined in terms of a repertoire of projection patterns (such as vestibulo-ocular, vestibulo-spinal and vestibulo-cerebellar) that are common to all vertebrates (*Pasqualetti et al., 2007*; *Straka and Baker, 2013*; *Straka et al., 2002*). However, neurons projecting to different targets are usually intermingled within a given nucleus or span more than one nucleus (*Büttner-Ennever, 1992*; *Daz and Puelles, 2003*; *Di Bonito et al., 2015*; *Pétursdóttir, 1990*). Accordingly, the different vestibular nuclei are comprised of highly heterogeneous neuronal populations. With the exception of the tangential nucleus, all the vestibular nuclei in birds have a homologous counterpart in vertebrates (*Straka and Baker, 2013*).

The Superior Vestibular Nucleus (VeS) is the most rostral of the vestibular nuclei and has a heterogeneous complement of cells (*Wold, 1976*). Electroporation with Atoh1-Cre + Flox-pA-GFP (Atoh1::GFP) resulted in scattered labelling of cells in the VeS, with diverse morphology (*Figure 4A*). Electroporation with Egr2-Cre + PBase + Pb-Flox-pA-GFP (Egr2::GFP) also labelled diverse VeS neurons (*Figure 4B*) that are most likely derived exclusively from r3, as r5-derived territory lies considerably caudal to the nucleus, with no labelled cells found in the intervening, presumably r4 derived, territory. Upon electroporation with Hoxb1-Cre + PBase + Pb-Flox-pA-GFP (Hoxb1::GFP), we observed a small number of isolated *Hoxb1*$^+$ cells in the VeS (*Figure 4C* - arrowheads). The rhombic lip and axial origin of VeS cells is summarised in *Figure 4D*.

The morphologically distinct Nucleus Deiters Ventralis (Dv) and Nucleus Deiters Dorsalis (Dd) are characterised by the presence of giant cells, amidst a collection of smaller cells of diverse morphologies (*Wold, 1976*). Electroporation with Atoh1::GFP resulted in abundant labelling of cells in the Dd and more scattered labelling in the Dv (*Figure 4E*). Giant, oval cells of the Dd were not observed labelled with Atoh1::GFP, while a single giant multipolar Dv cell was labelled with Atoh1::GFP in only one occasion, in contrast with more frequent labelling of smaller cells (*Figure 4F*). Electroporation with Ptf1a-Cre + PBase + Pb-Flox-pA-GFP (Ptf1a::GFP) also resulted in abundant labelling of cells in the Dd and Dv (*Figure 4G*), including numerous small cells (*Figure 4H* – top panel,

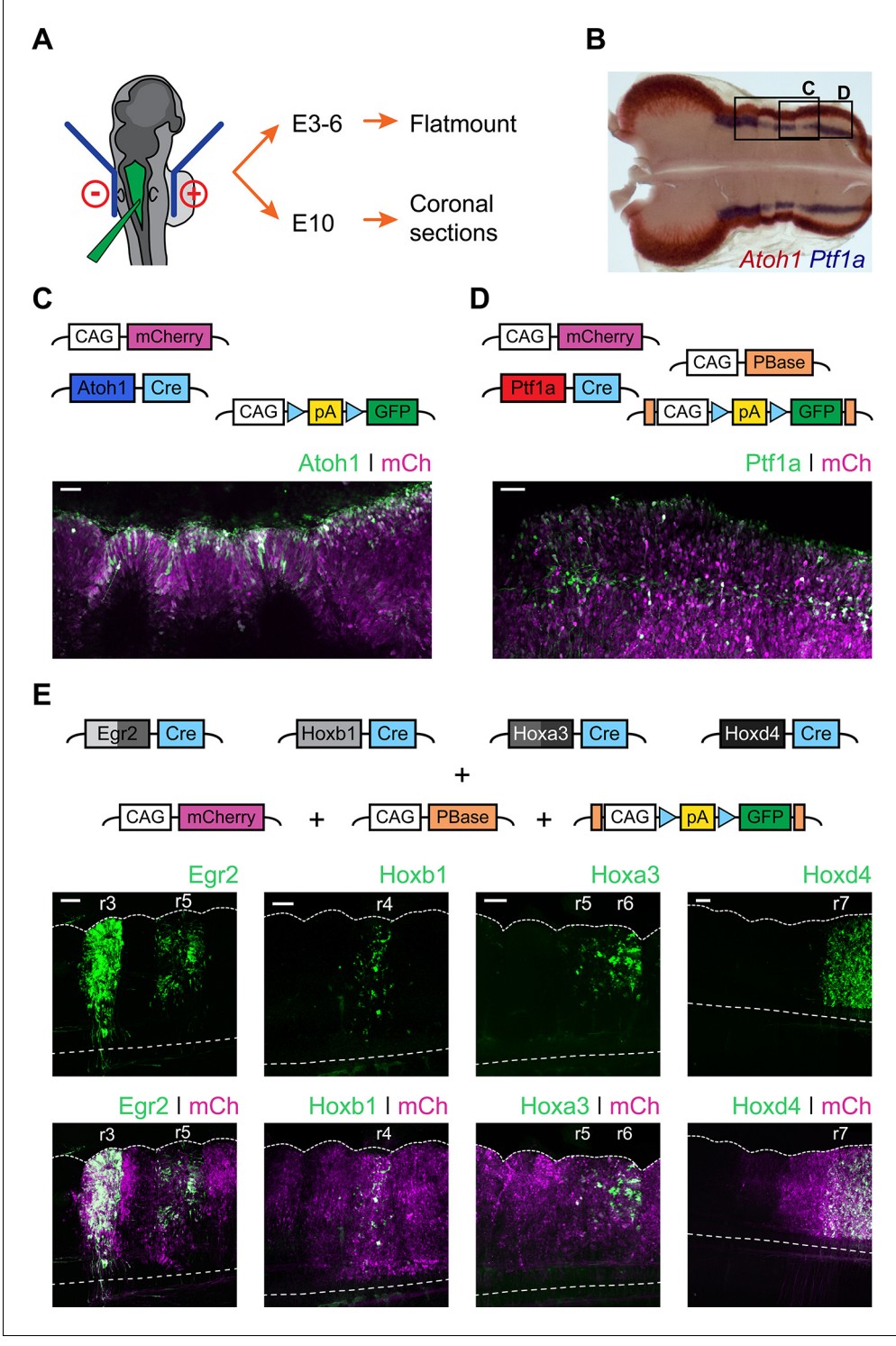

**Figure 2.** Fate mapping the developmental origin of hindbrain auditory and vestibular nuclei. (**A**) Diagram of the electroporation strategy. (**B**) Representative flatmount view of a HH21 chick hindbrain stained by in situ hybridisation for *Atoh1* (red) and *Ptf1a* (blue). (**C-D**) *Top panels.* Plasmid constructs employed for labelling *Atoh1*[+] (**C**) and *Ptf1a*[+] (**D**) progenitors. *Bottom panels.* Representative close up flatmount image of HH17 chick hindbrains, electroporated at HH14 with Atoh1-Cre + CAG-Flox-pA-GFP + CAG-mCherry (**C**) or Ptf1a-Cre + CAG-PBase + Pb-CAG-Flox-pA-GFP + CAG-mCherry (**D**). Scale bar: 50 µm. (**E**) *Top panels.* Plasmid constructs employed for labelling cells arising from specific rhombomeres. *Bottom panels.* Representative close up flatmount images of HH17 (HH28 for *Hoxd4*) chick hindbrains, electroporated at HH14 with CAG-PBase +Pb-CAG-Flox-pA-

*Figure 2 continued on next page*

*Figure 2 continued*

GFP + CAG-mCherry and Egr2-Cre, Hoxb1-Cre, Hoxa3-Cre or Hoxd4-Cre, from left to right, respectively. Dotted lines outline the lateral border and the ventral midline. Scale bar: 100 μm.

DOI: https://doi.org/10.7554/eLife.40232.003

arrowheads), that showed a different morphology to the characteristic Dd giant oval cells (*Figure 4H* – top panel, asterisks), and medium/small size cells in the Dv (*Figure 4H* – bottom panel, arrowheads).

Axial markers allocated the origin of the Deiters nuclei to the middle hindbrain rhombomeres. Electroporation with Egr2::GFP resulted in scattered labelling of Dd (*Figure 4I* – left panel) and Dv (*Figure 4I* – right panel) cells, none of which showed morphological features of giant cells. Electroporation with Hoxb1:: GFP also resulted in labelling of cells in the Dd and Dv, showing a wider range of morphologies, including giant oval cells in the Dd (*Figure 4J* – left panel) and giant multipolar cells in the Dv (*Figure 4J* – right panel). In summary, both *Atoh1* and *Ptf1a* progenitors from r3-r5 gave rise to cells in the dorsal and ventral Deiter's nuclei (*Figure 4D*). The distinct giant cells appeared to originate from r4 progenitors that may not correspond to either *Atoh1*$^+$ or *Ptf1a*$^+$ lineages.

The Nucleus Tangentialis (Ta) is composed of elongated cells that are located tangential to the incoming VIIIth nerve fibres (*Wold, 1976*). Electroporation with Atoh1::GFP did not label cells in the Ta (*Figure 4K*), while Ptf1a::GFP labelling showed only scattered small cells in the Ta territory, none of which corresponded to the characteristic elongated Ta cells (*Figure 4L*). The latter were labelled by electroporation with Egr2::GFP (*Figure 4M*), Hoxb1::GFP (*Figure 4N*) and Hoxa3-Cre + PBase + Pb-Flox-pA-GFP (Hoxa3::GFP) (*Figure 4O*), indicating they may derive from *Atoh1*$^-$/*Ptf1a*$^-$negative precursors from r4-r6 (*Figure 4D*).

The Medial Vestibular Nucleus (VeM) is located beneath the ventricular surface and closely associated with the auditory NM. Contralaterally projecting NM axons divide the VeM into dorsomedial and ventrolateral parts (*Figure 5A*). Electroporations with Atoh1::GFP showed scattered labelling of VeM cells (*Figure 5A* and inset), including neurons embedded within the NM fibre tract (*Figure 5A* – arrow). Electroporation with Ptf1a::GFP revealed abundant *Ptf1a* derived neurons in both parts of the nucleus (*Figure 5B*). The VeM has the longest rostrocaudal extent of all the vestibular nuclei (*Wold, 1976*). Egr2::GFP electroporations labelled numerous cells in the rostral VeM (*Figure 5C*- left panel), and only scattered cells more caudally, at the level adjacent to the rostral NM (*Figure 5C* – right panel, arrowheads). VeM cells at this axial level were also labelled via electroporation with *Hoxb1* (*Figure 5D*) and *Hoxa3* (*Figure 5E*) reporter constructs, whilst the middle and caudal VeM was mainly labelled by Hoxd4-Cre + PBase + Pb-Flox-pA-GFP (Hoxd4::GFP) electroporations (*Figure 5F*). In summary, VeM cells were observed originating from *Atoh1*$^+$ and *Ptf1a*$^+$ progenitors spanning from r3 to axial levels caudal to r7 (*Figure 5G*).

The Descending Vestibular Nucleus (VeD) extends rostrocaudally from the level of the VIIIth nerve root to the caudal end of the hindbrain (*Wold, 1976*). Electroporation with Atoh1::GFP resulted in profuse labelling of neurons in the caudal VeD (*Figure 5H*) and more scattered cell labelling in the rostral portion of the nucleus, through which NL axons course into the ventral hindbrain and SON axons reach the NL (*Lachica et al., 1994*; *Takahashi and Konishi, 1988*) (*Figure 5I*). Electroporations with Ptf1a::GFP resulted in labelling of small cells throughout the VeD (*Figure 5J*). The rostral portion of the VeD showed cells labelled via electroporation with Egr2::GFP (*Figure 5K*) and Hoxa3::GFP (*Figure 5L*). In turn, the middle and caudal portions of the VeD were labelled by electroporation with Hoxd4::GFP (*Figure 5M*). Finally, electroporations with Hoxb1::GFP labelled scattered cells in the rostral VeD (*Figure 5N*), which may have migrated caudally from r4. In summary, our characterisation of the VeD showed cells arising from both *Atoh1* and *Ptf1a* progenitors that span r4 to levels caudal to r7 (*Figure 5G*).

## Avian hindbrain first order auditory nuclei are defined by axially distinct pools of *Atoh1* and *Ptf1a* precursors

In birds, primary auditory afferents synapse at the Nucleus Magnocellularis (NM) and Nucleus Angularis (NA) (*Ryugo and Parks, 2003*). NM neurons, characterised by their bifurcating ipsilateral and

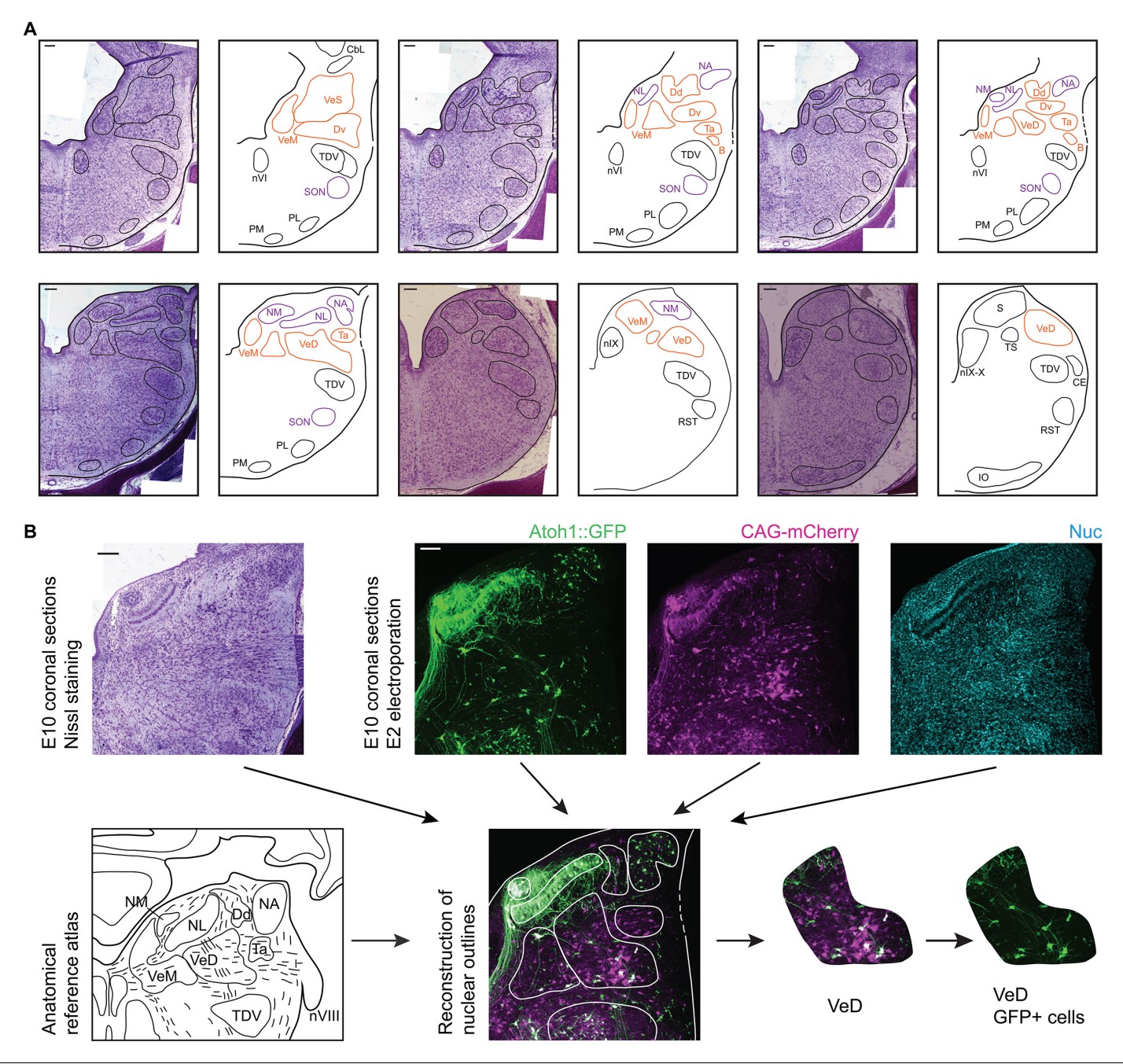

**Figure 3.** Anatomy of hindbrain auditory and vestibular nuclei. (**A**) Representative images of coronal sections of Nissl stained E10 chick hindbrain (left panels) and diagrams (right panels), showing the outlines of hindbrain nuclei. From left to right and top to bottom: rostral to caudal. Scale bars: 50 μm. All drawings are to scale. B, Cell group B; Cbl, Nucleus Cerebellaris Internus; CE, Nucleus Cuneatus Externus; Dd, Nucleus Deiters Dorsalis; Dv, Nucleus Deiters Ventralis; IO, Inferior Olive Nucleus; NA, Nucleus Angularis; nIX, Glossopharyngeal Nucleus; nIX-X, Glossopharyngei and Vagus Nucleus; NL, Nucleus Laminaris; NM, Nucleus Magnocellularis; nVI, Abducens Nucleus; PL, Lateral Pontine Nucleus; PM, Medial Pontine Nucleus; RST, Reticular Subtrigeminal Nucleus; S, Solitaris Nucleus; SON, Superior Olivary Nucleus; Ta, Tangential Nucleus; TDV, Descending Trigeminal Nucleus; TS, Torus Semicircularis; VeD, Descending Vestibular Nucleus; VeM, Medial Vestibular Nucleus; VeS, Superior Vestibular Nucleus. (**B**) Example of a nucleus identification by combining information from Nissl stained sections (top left), fluorescent protein expression and nuclear staining (top right) and anatomical reference atlases (bottom left, modified from (***Kuenzel and Masson, 1988***; ***Wold, 1976***)).
DOI: https://doi.org/10.7554/eLife.40232.004

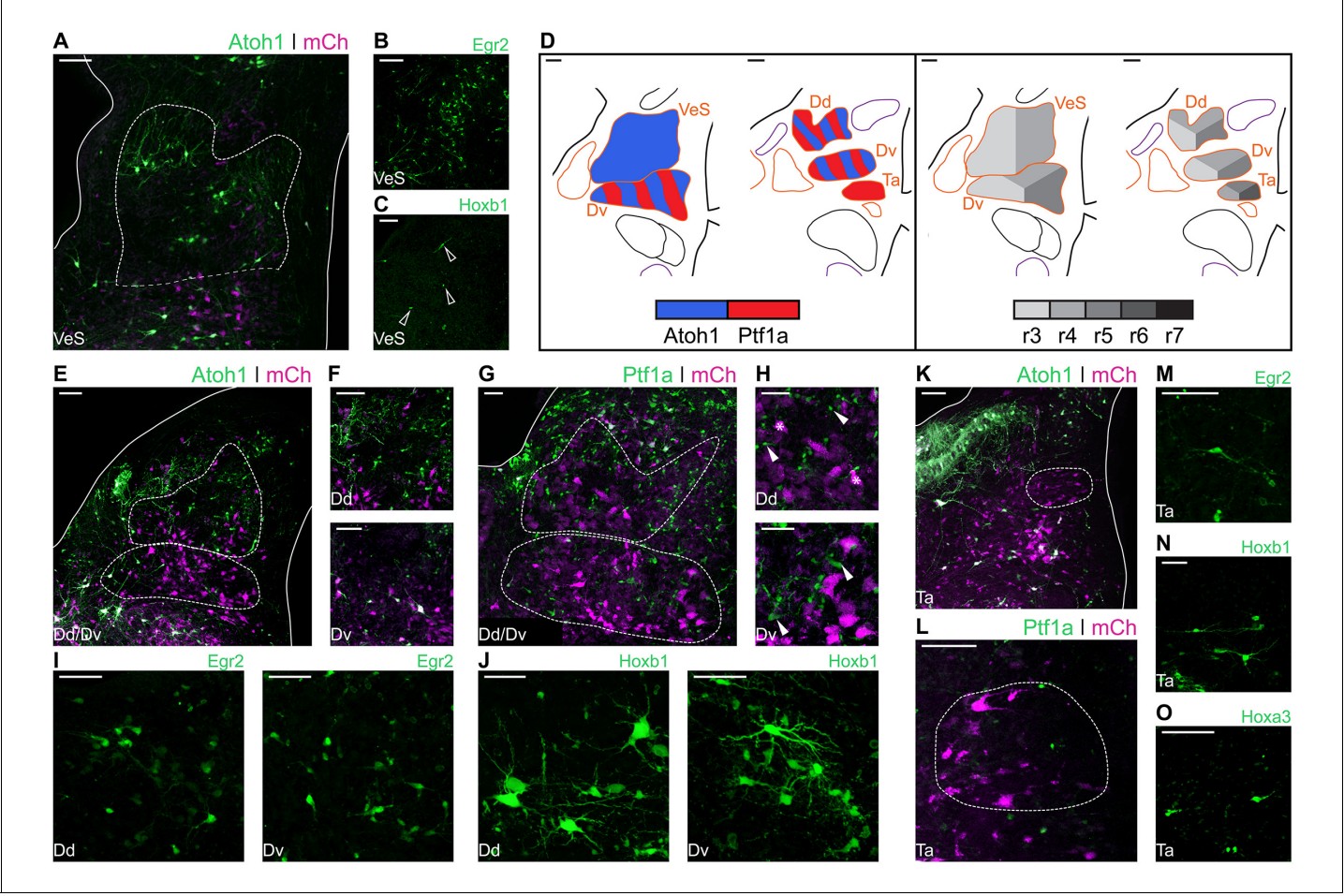

**Figure 4.** Developmental origin of avian Superior, Deiters and Tangential vestibular nuclei. (**A**) Close up view of the VeS from an E10 chick hindbrain coronal section electroporated at HH14 with Atoh1::GFP and CAG-mCherry. Solid lines, hindbrain borders. Dotted lines, VeS borders. Scale bar: 100 µm. (**B-C**) Close up view of the VeS from an E10 chick hindbrain coronal section electroporated at HH14 with Egr2::GFP (**B**) or Hoxb1::GFP (**C**). Scale bars: 100 µm. (**D**) Diagrams summarising the contributions of $Atoh1^+/Ptf1a^+$ progenitors (left panel) or different rhombomeres (right panel) to the superior, deiters and tangential vestibular nuclei. Scale bars: 50 µm. (**E**) Close up view of the Dd and Dv from an E10 chick hindbrain coronal section electroporated at HH14 with Atoh1::GFP and CAG-mCherry. Solid lines, hindbrain borders. Dotted lines, Dd and Dv borders. Scale bar: 100 µm. (**F**) Detailed view of $Atoh1^+$ cells in the Dd (top panel) and the Dv (bottom panel). Scale bars: 100 µm. (**G**) Close up view of the Dd and Dv from an E10 chick hindbrain coronal section electroporated at HH14 with Ptf1a::GFP and CAG-mCherry. Solid line, hindbrain borders. Dotted lines, Dd and Dv borders. Scale bar: 50 µm. (**H**) Detailed view of $Ptf1a^+$ cells in the Dd (top panel) and the Dv (bottom panel). Scale bars: 50 µm. (**I-J**) Detailed view of $Egr2^+$ (**I**) and $Hoxb1^+$ (**J**) cells in the Dd (left panel) and the Dv (right panel). Scale bars: 50 µm. (**K**) Close up view of the Ta from an E10 chick hindbrain coronal section electroporated at HH14 with Atoh1::GFP and CAG-mCherry. Solid lines, hindbrain borders. Dotted line, Ta borders. Scale bar: 100 µm. (**L**) Close up view of the Ta from an E10 chick hindbrain coronal section electroporated at HH14 with Ptf1a::GFP and CAG-mCherry. Scale bar: 100 µm. (**M-O**) Detailed view of $Egr2^+$ (**M**), $Hoxb1^+$ (**N**), $Hoxa3^+$ (**O**) cells in the Ta. Scale bars: 50, 100 and 100 µm, respectively.

DOI: https://doi.org/10.7554/eLife.40232.005

contralateral projections, are labelled by electroporation with Atoh1::GFP (**Figure 6A**). Ipsilateral axonal projections from NM neurons were observed surrounding the lateral border of the NM on their way towards the Nucleus Laminaris (NL) and terminating on its dorsal region (**Figure 6B**). At the contralateral (non-electroporated) side, NM axons were observed contacting the ventral side of NL neurons (**Figure 6C**). At its caudal end, the border between NM and the Descending Vestibular Nucleus (VeD) is difficult to distinguish (**Figure 6D**). No $Ptf1a$ labelled neurons were found within the NM upon electroporation with Ptf1a::GFP (**Figure 6E**).

Electroporation with Egr2::GFP (**Figure 6F**) and Hoxa3::GFP (**Figure 6G**) resulted in scattered labelling of NM cells, arising from r5 and r6. Electroporation with Hoxb1::GFP labelled cells in the NM area, but not $Atoh1^+$ NM neurons, labelled with Atoh1-Gal4 + UAS-tdT (Atoh1::tdT)

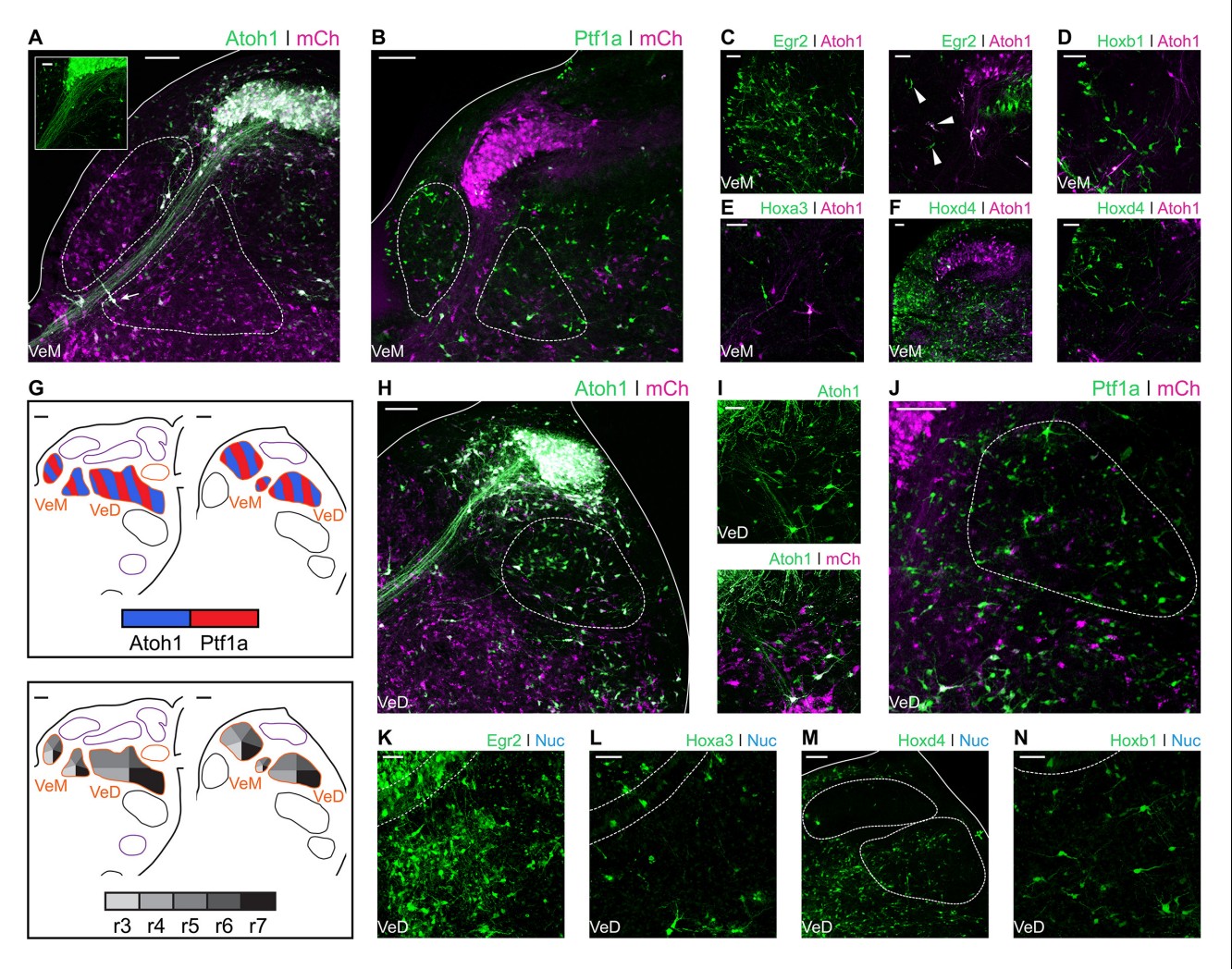

**Figure 5.** Developmental origin of avian Medial and Descending vestibular nuclei. (**A**) Close up view of the VeM from an E10 chick hindbrain coronal section electroporated at HH14 with Atoh1::GFP and CAG-mCherry. Scale bar: 100 µm. Inset: detailed view of VeM *Atoh1+* cells. Solid line, hindbrain border. Dotted lines, VeM borders. Scale bar: 50 µm. (**B**) Close up view of the VeM from an E10 chick hindbrain coronal section electroporated at HH14 with Ptf1a::GFP and CAG-mCherry. Solid line, hindbrain border. Dotted lines, VeM borders. Scale bar: 100 µm. (**C-F**) Close up view of the VeM from E10 chick hindbrain coronal sections electroporated at HH14 with Atoh1-Gal4 + UAStdT and Egr2::GFP (**C**. left panel, rostral. Right panel, caudal), Hoxb1:: GFP (**D**), Hoxa3::GFP (**E**) or Hoxd4::GFP (**F** left panel, rostral. Right panel, caudal). Scale bars: 50 µm. (**G**) Diagrams summarising the contributions of *Atoh1+/Ptf1a+* progenitors (left panel) or different rhombomeres (right panel) to the medial and descending vestibular nuclei. Scale bars: 50 µm. (**H**) Close up view of the VeD from an E10 chick hindbrain coronal section electroporated at HH14 with Atoh1::GFP and CAG-mCherry. Scale bar: 100 µm. (**I**) Detailed view of *Atoh1+* cells in the rostral VeD. Solid lines, hindbrain border. Dotted line, VeD border. Scale bar: 50 um. (**J**) Close up view of the VeD from an E10 chick hindbrain coronal section electroporated at HH14 with Ptf1a::GFP and CAG-mCherry. Dotted line, VeD border. Scale bar: 100 µm. (**K-N**) Close up view of the VeD from E10 chick hindbrain coronal sections electroporated at HH14 with Egr2::GFP (**K**), Hoxa3::GFP (**L**), Hoxd4::GFP (**M**) or Hoxb1::GFP (**N**). Solid lines, hindbrain border (**M**). Dotted lines, NL borders (**K, L and N**), NM and VeD borders (**M**). Scale bars: 50 µm.

DOI: https://doi.org/10.7554/eLife.40232.006

(*Figure 6H*). Finally, electroporation with Hoxd4::GFP resulted in abundant labelling of NM cells (*Figure 6I*). This caudal rhombic lip origin of NM cells was confirmed via a transectional labelling strategy, wherein electroporation with Hoxd4-Cre + Atoh1-FLPo + Flox/FRT-Flox-pA-GFP + CAG-mCherry resulted in GFP expression after double Cre/Flp recombination driven by *Hoxd4* and *Atoh1* enhancer elements (*Figure 6J*). The rhombic lip and axial origin of NM cells is summarised in *Figure 6K*.

The Nucleus Angularis (NA) is composed of at least four morphologically and electrophysiologically distinct neuronal types: radial, vertical, stubby and planar cells (*Soares and Carr, 2001*;

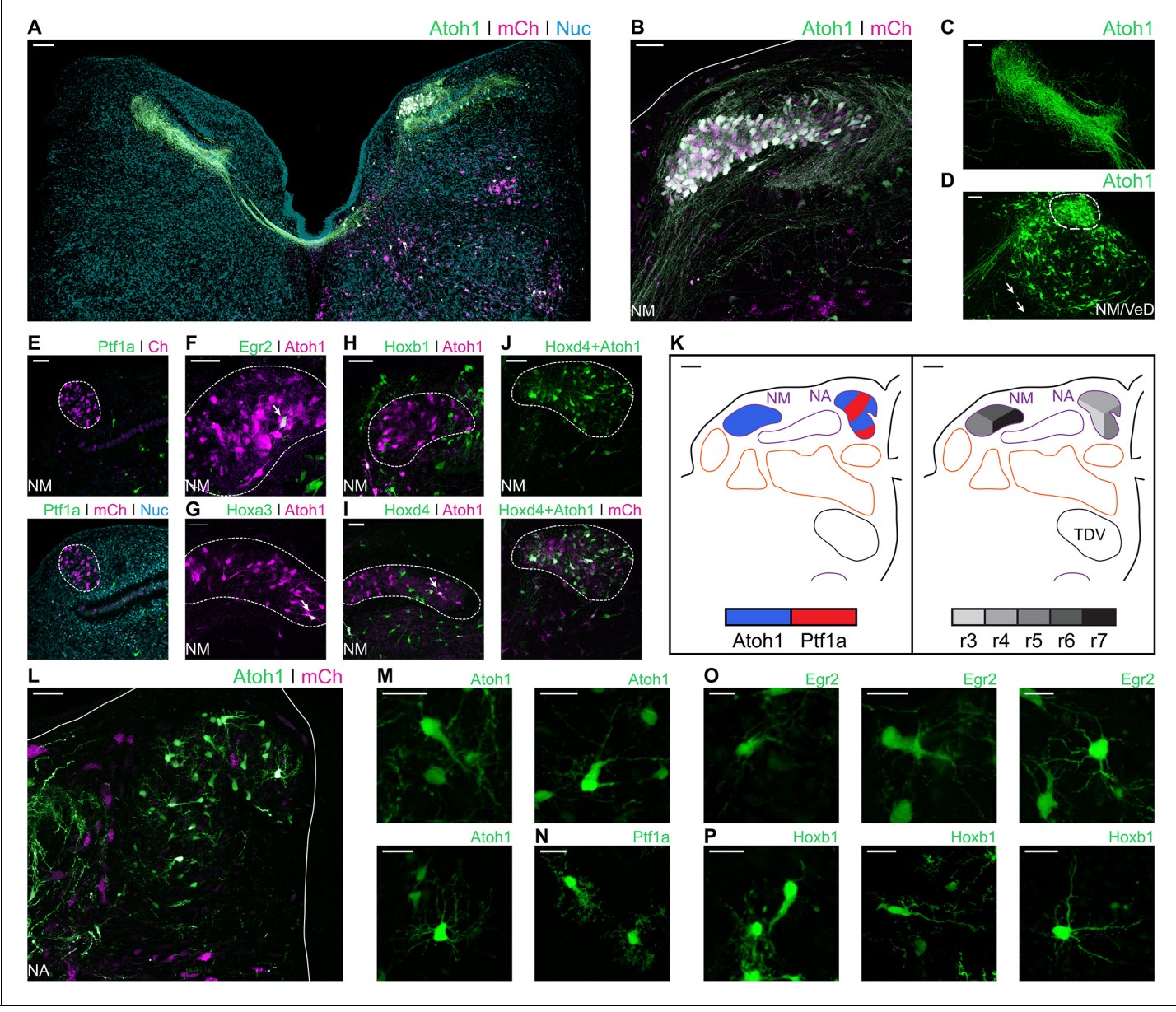

**Figure 6.** Developmental origin of chick first order hindbrain auditory nuclei. (A) Representative image of an E10 chick hindbrain coronal section electroporated at HH14 with Atoh1-Cre + CAG-Flox-pA-GFP (Atoh1::GFP) and CAG-mCherry. Scale bar: 100 μm. (B) Close up view of the NM from a section caudal to the one shown in A. Solid line, hindbrain border. Scale bar: 50 μm. (C) Close up view of *Atoh1*[+] NM projections to the contralateral NL. Scale bar: 50 μm. (D) Close up view of the caudal NM showing a diffuse boundary with the VeD. Dotted line shows NM border. Arrows show VeD projections that join the contralateral NM axons. Scale bar: 50 μm. (E) Close up view of the NM from an E10 chick hindbrain coronal section electroporated at HH14 with Ptf1a-Cre, PBase, Pb-CAG-Flox-pA-GFP (Ptf1a::GFP) and CAG-mCherry. Dotted line shows NM border. Scale bar: 50 μm. (F-I) Close up view of the NM from E10 chick hindbrain coronal sections electroporated at HH14 with CAG-PBase + Pb-CAG-Flox-pA-GFP + Atoh1-Gal4 + UAS-tdT (Atoh1::tdT) and Egr2-Cre (F), Hoxa3-Cre (G), Hoxb1-Cre (H) or Hoxd4-Cre (I). Scale bars: 50 μm. Arrows point to *Egr2*[+]/*Atoh1*[+] (F), *Hoxa3*[+]/*Atoh1*[+] (G) and *Hoxd4*[+]/*Atoh1*[+] (I) cells. (J) Close up view of the NM from an E10 chick hindbrain coronal section electroporated at HH14 with Hoxd4-Cre + Atoh1-FLPo + CAG-Flox-FLp-pA-GFP + CAG-mCherry. Dotted line shows NM border. Scale bar: 50 μm. (K) Diagrams summarising the contributions of *Atoh1*[+]/*Ptf1a*[+] progenitors (left panel) or different rhombomeres (right panel) to the first order auditory nuclei. Scale bars: 50 μm. (L) Close up view of the NA from an E10 chick hindbrain coronal section electroporated at HH14 with Atoh1::GFP and CAG-mCherry. Solid lines, hindbrain borders. Scale bar: 50 μm. (M) Close up views of representative vertical (top-left panel), planar (top-right panel) and radial (bottom panel) NA *Atoh1*[+] cells. Scale bars: 25 μm. (N) Close up view of representative stubby NA *Ptf1a*[+] cells. Scale bar: 20 μm. (O) Close up views of representative vertical, (left panel), planar (middle panel) and radial (right panel) NA *Egr2*[+] cells. Scale bars: 20 μm. (P) Close up views of representative vertical and stubby, (left panel), planar (middle panel) and radial (right panel) NA *Hoxb1*[+] cells. Scale bars: 20 μm.

DOI: https://doi.org/10.7554/eLife.40232.007

*Soares et al., 2002*). Electroporation with Atoh1::GFP resulted in extensive labelling of NA neurons (*Figure 6L*), that showed morphologies indicative of radial, vertical and planar cells (*Figure 6M*). Electroporation with Ptf1a::GFP resulted in more scattered labelling of the NA. *Ptf1a*+ NA cells had the distinctive spiny morphology of stubby cells (*Figure 6N*). Planar, radial and vertical cells (likely *Atoh1*), but not stubby (likely *Ptf1a*) cells, were labelled following electroporation with Egr2::GFP (*Figure 6O*). By contrast, all four cell types were labelled by Hoxb1::GFP electroporations (*Figure 6P*). This suggests that, while *Atoh1*+ NA cells derived from r3, r4 and r5 (and this may include vertical, planar and radial morphological types), only r4 *Ptf1a*+ progenitors gave rise to NA stubby cells (*Figure 4K*).

In summary, we observed that the *Atoh1* and *Ptf1a* electroporations labelled all the described cell types in the avian first order hindbrain auditory nuclei. While the axial origin of NA cells was located towards the more rostral rhombomeres, NM cells were observed arising from the caudal rhombic lip (*Figure 6K*).

## The *Atoh1* positive rhombic lip is a main source of avian hindbrain second order auditory neurons

In chick, the Nucleus Laminaris (NL) and Superior Olivary Nucleus (SON) comprise the hindbrain second order neuronal populations. They receive input from the NA and NM and are involved in the processing of monaural and binaural auditory cues (*Carr and Soares, 2002*). The NL sits ventral to the NM and is composed of neurons aligned on a plane along the mediolateral axis. Electroporation with Atoh1::GFP resulted in strong labelling of NL neurons (*Figure 7A*) that reveals their characteristic bipolar dendritic morphology (*Figure 7A* - **inset**), with segregated dorsal and ventral dendrites.

Electroporations with Egr2::GFP resulted in the two distinct bands of GFP expression corresponding to r3 and r5. Cells of the NL were observed in the caudal band of GFP expression (*Figure 7B*) suggesting they originated from progenitors in r5, but not r3. NL cells were also labelled by electroporation with Hoxb1::GFP (*Figure 7C*) and Hoxa3::GFP (*Figure 7D*), indicating NL neurons derived from r4-r6 progenitors as summarised in *Figure 7E*. No *Ptf1a* labelled neurons were found within the NL.

The SON is located in the ventrolateral hindbrain (*Figures 3* and *7F*). Electroporation with Atoh1::GFP resulted in extensive labelling of SON neurons (*Figure 7F*). By contrast, electroporation with Ptf1a::GFP only occasionally labelled one or two small cells in the nucleus (*Figure 7G* - **arrowhead**).

Electroporation with Egr2::GFP resulted in extensive labelling of SON neurons (*Figure 7H*). Co-labelling with Egr2::GFP and Atoh1::tdT showed both *Egr2*+/*Atoh1*- and *Egr2*-/*Atoh1*+ cells, as well as many double labelled *Egr2*+/*Atoh1*+ cells (*Figure 7I*). This suggested that, within r5, *Atoh1*+ progenitors are not the exclusive source of SON neurons. In addition, electroporation with Hoxb1::GFP (*Figure 7J*) and Hoxa3::GFP (*Figure 7K*) labelled scattered cells in the SON, suggesting a small contribution from both r4 and r6 to the predominantly r5 derived SON (*Figure 7E*).

In summary, both second order auditory nuclei originated from mid-hindbrain progenitors, namely, r4 to r6. The NL, composed of only one cell type, was labelled by *Atoh1* electroporation, indicating that it is derived from rhombic lip progenitors. In contrast, the SON is a more heterogeneous nucleus. Accordingly, we observed cells that originated from both *Atoh1*+ and *Atoh1*- progenitors. While the latter may include cells of *Ptf1a* origin, these most likely originated from other progenitor populations.

## Discussion

In this study, we have mapped the rhombic lip/*Atoh1*, ventricular zone/*Ptf1a* and axial origin of chick hindbrain vestibular and auditory nuclei. This shows, for the first time, the origins of both systems by rhombomeric identity and specific dorsoventral lineage. Our data give a solid framework by which to assess divergence and convergence within hindbrain functional circuits in birds and mammals. We evaluate the contribution of changes in hindbrain development to the evolution of neuronal circuits, within the context of disparaging evolutionary histories of the different amniote vestibular and auditory structures.

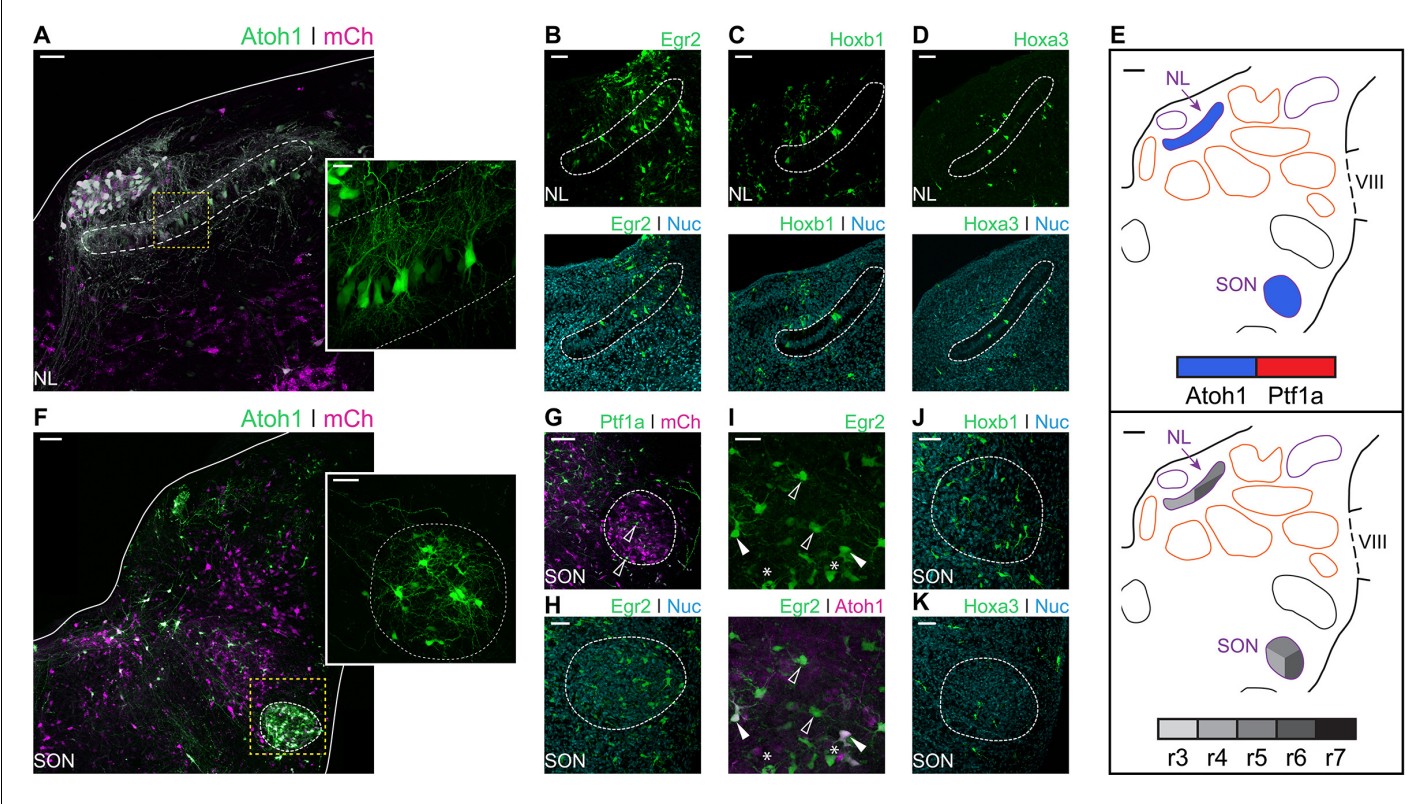

**Figure 7.** Developmental origin of avian second order hindbrain auditory nuclei. (**A**) Close up view of the NL from an E10 chick hindbrain coronal section electroporated at HH14 with Atoh1::GFP and CAG-mCherry. Solid line, dorsal border of the hindbrain. Dotted line, NL border. Yellow dotted square, approximate location of inset image. Scale bar: 50 µm. Inset: detailed view of *Atoh1*+ NL cells depicting bipolar morphology. Scale bar: 20 µm. (**B-D**) Close up view of the NL from E10 chick hindbrain coronal sections electroporated at HH14 with Egr2::GFP (**B**), Hoxb1::GFP (**C**), Hoxa3::GFP (**D**) and counterstained with NucRed. Dotted lines show NL borders. Scale bars: 50 µm. (**E**) Diagrams summarising the contributions of *Atoh1*+/*Ptf1a*+ progenitors (top panel) or different rhombomeres (bottom panel) to the second order auditory nuclei. Scale bars: 50 µm. (**F**) Close up view of the SON from an E10 chick hindbrain coronal section electroporated at HH14 with Atoh1::GFP and CAG-mCherry. Solid lines, hindbrain borders. Dotted line, SON border. Yellow dotted square, approximate location of inset image. Scale bar: 100 µm. Inset: detailed view of *Atoh1*+ SON. Scale bar: 50 µm. (**G**) Close up view of the SON from an E10 chick hindbrain coronal section electroporated at HH14 with Ptf1a::GFP and CAG-mCherry. Dotted line shows SON border. Scale bar: 100 µm. (**H**) Close up view of the SON from an E10 chick hindbrain coronal section electroporated at HH14 with Egr2::GFP and counterstained with NucRed. Dotted line shows SON border. Scale bar: 50 µm. (**I**) Close up view of the SON from an E10 chick hindbrain coronal section electroporated at HH14 with Egr2::GFP and Atoh1::tdT. Filled arrowheads show *Egr2*+/*Atoh1*+ cells; empty arrowheads show *Egr2*+/*Atoh1*- cells; asterisks show *Egr2*-/*Atoh1*+ cells. Scale bar: 50 µm. (**J-K**. Close up view of the SON from E10 chick hindbrain coronal sections electroporated at HH14 with Hoxb1::GFP (**J**) or Hoxa3::GFP (**K**) and counterstained with NucRed. Dotted lines show SON borders. Scale bars: 50 µm.

DOI: https://doi.org/10.7554/eLife.40232.008

## Anatomical, morphological and developmental features indicate a high degree of conservation of hindbrain vestibular nuclei across vertebrates

Vestibular peripheral organs, hindbrain vestibular nuclei and their connectivity patterns are highly conserved across vertebrates and clear evolutionary relationships can be established (*Straka and Baker, 2013*). The axial developmental origin and location of vestibular neurons projecting to oculo-motor, spinal cord, cerebellar and commissural targets are also conserved (*Branoner et al., 2016*; *Chagnaud et al., 2017*; *Malinvaud et al., 2010*; *Straka and Baker, 2013*; *Straka et al., 2014*), however, a comparative analysis of the dorsoventral lineage of hindbrain vestibular neurons has been lacking. Here, we show that in chick, cells derived from rhombic lip *Atoh1*+ progenitors were present in the Superior, Deiters, Medial and Descending vestibular nuclei, while we observed no contribution from this lineage to the avian exclusive Tangential nucleus (*Figures 4* and *5*). Moreover, we have identified cells derived from *Ptf1a*+ progenitors in the Deiters, tangential, medial and descending

vestibular nuclei (*Figures 4* and *5*). These results are similar to fate maps in mouse showing that rhombic lip *Atoh1*[+] progenitors contribute to the superior (*Wang et al., 2005*), lateral, medial and spinal vestibular nuclei (*Rose et al., 2009*), while *Ptf1a*[+] progenitors give rise to vestibular hindbrain neurons, possibly located in all four nuclei (*Yamada et al., 2007*). Of note, *Atoh1* and *Ptf1a* cells do not account for the whole diversity of cell types, in either chick or mouse (*Figure 8*), and further mapping efforts are required to fully characterise the genetic lineages that give rise to the myriad of hindbrain vestibular neurons.

Our high resolution electroporation-based labelling allowed for a detailed characterisation of the axial origin of vestibular neuronal types (*Figure 8*). The Nucleus Deiters Ventralis (Dv) and Nucleus Deiters Dorsalis (Dd) of birds are homologous to the rostroventral and dorsocaudal part of the mammalian lateral vestibular nucleus, respectively (*Passetto et al., 2008*), which is mainly derived from r4 progenitors (*Chen et al., 2012*; *Di Bonito et al., 2015*), with additional contribution from r3 and r5 (*Pasqualetti et al., 2007*). Here, we observed a similar axial origin for both the Dv and Dd, with progenitors allocated to r3, r4 and r5 (*Figures 4I,J* and *8*). Moreover, Dv and Dd giant cells derived from *Hoxb1*[+] (r4) progenitors. These may constitute a homologous cell type to the r4 derived large stellate cells of the murine lateral (and medial) vestibulospinal tract (*Di Bonito et al., 2015*; *Di Bonito et al., 2017*). Finally, we observed a minor group of small r4 derived cells that integrate into the VeS and VeM, invading r3 and r5 derived territories, respectively (*Figures 4B* and *5* D). A similar group of cells has been described in mouse (*Di Bonito et al., 2017*) and shown to also belong to the highly conserved vestibulospinal tract. Taken together, these observations support a homologous axial and dorso-ventral origin for spinal cord projecting neurons from both species and highlight the high degree of conservation in the detailed organisation of the ancestral vestibulospinal tract (*Straka and Baker, 2013*).

## Avian first order hindbrain auditory neurons may be related to vertebrate ancestral vestibuloacoustic neurons

Our electroporation-based fate mapping showed neurons in the chick first order NA arising from either *Atoh1* or *Ptf1a* progenitors (*Figures 6K–N* and *8*), thus sharing a developmental origin with the neurons of the mammalian ventral cochlear nucleus (VCN) (*Figure 8* and (*Fujiyama et al., 2009*)). Moreover, both NA and VCN neurons share a common axial origin from rostral rhombomeres (*Figures 6K,O–P* and *8* and (*Cramer et al., 2000*; *Di Bonito et al., 2013*; *Di Bonito et al., 2017*; *Farago et al., 2006*; *Marín and Puelles, 1995*)). This developmental conservation suggests that anterior neurons of the ascending auditory pathway of amniotes may be homologous. Furthermore, first order NA neurons in birds/reptiles were proposed to be related to dorsal neurons of the fish anterior octaval nucleus (*Walton et al., 2017*). These anteriorly derived neurons may thus comprise a developmentally conserved brainstem first order relay point for auditory (particle vibration) information.

The first order NM occupies a major portion of the diapsid caudal/dorsal hindbrain. NM neurons are derived from progenitors in the alar plate, the dorsal half of the neuroepithelium (*Tan and Le Douarin, 1991*) of the caudal hindbrain (*Cambronero and Puelles, 2000*; *Cramer et al., 2000*; *Marín and Puelles, 1995*). Our enhancer-based mapping confirmed this and further showed that NM neurons originate exclusively from rhombic lip (*Atoh1*[+]) progenitors located caudal to r5 (*Figures 6A–K* and *8*). In zebrafish, *Atoh1* is required for the development of a subpopulation of *zn-5*[+] and *Lhx2/9*[+] cells that may correspond to contralaterally projecting octaval neurons (*Sassa et al., 2007*) and preliminary mapping data shows *Atoh1* labelling of neurons on the caudal octaval nucleus (*Wullimann et al., 2011*). Moreover, it has been proposed that NM neurons may derive from an ancestral population equivalent to extant neurons of the dorsal descending octaval nucleus of fish (*Carr and Christensen-Dalsgaard, 2016*; *Walton et al., 2017*), which are part of a binaural circuit that sharpens directional information from the saccule (*Edds-Walton, 2016*). Overall, this suggests that extant caudal octaval (vestibuloacoustic) neurons in chick that belong to the auditory NM and the vestibular DeV may be ancestrally related. Morphologically, a close association was observed between the two nuclei at the level of the caudal end of the NM (*Figure 6D*) with some VeD cells extending contralateral projections that join the NM dorsal (cochlear) commissure (*Figure 6D* - arrows). Finally, our observation that NM and VeD neurons share a developmental origin in the caudal rhombic lip, with a significant number of VeD neurons labelled via *Atoh1* electroporations (*Figure 5H*) supports a common evolutionary origin for NM and rhombic lip derived VeD neurons.

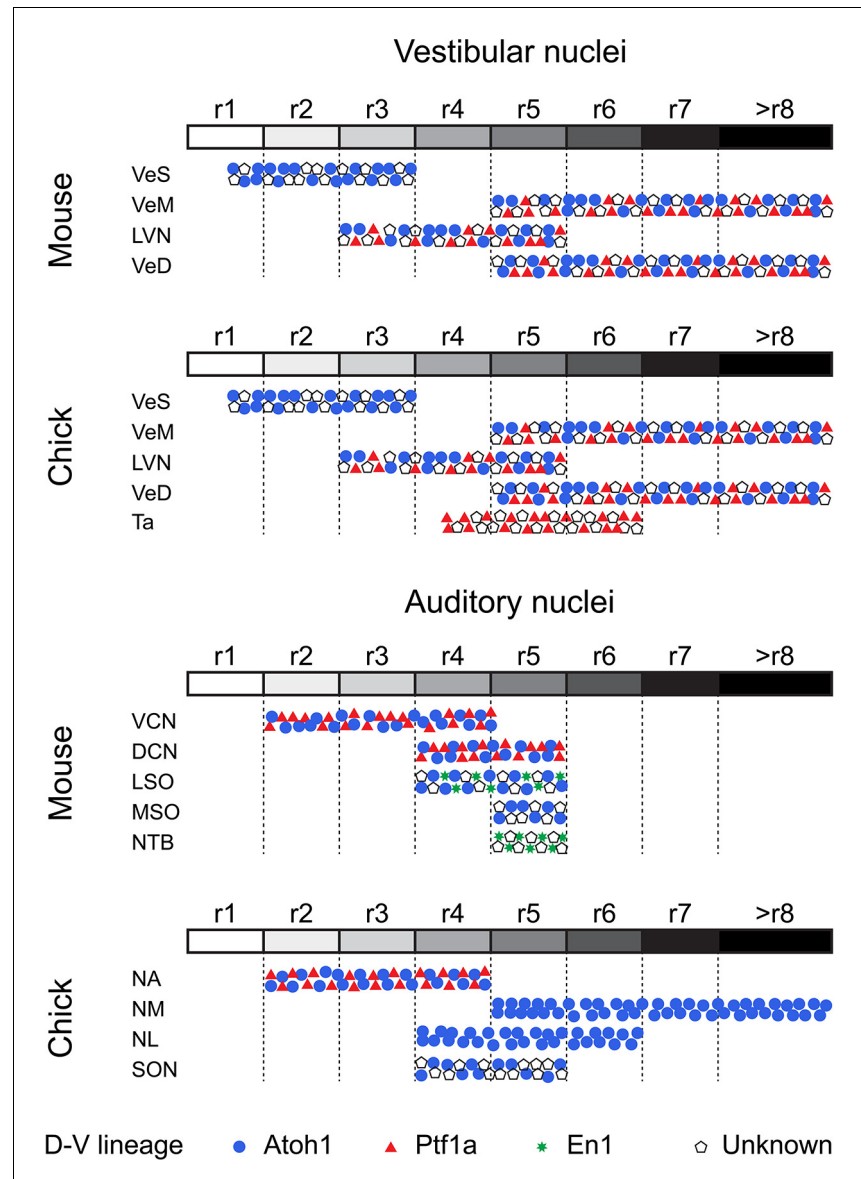

**Figure 8.** Comparative developmental origin of mammalian and avian vestibular and auditory hindbrain nuclei. Schematic diagram showing the lineage origin of cells in the auditory and vestibular nuclei (blue circles, *Atoh1*; red triangles, *Ptf1a*; green stars, *En-1*; empty pentagons, unknown), alongside their rhombomeric origin (left to right, r1 to >r8, white to black grey scale). Chick mapping data, this work and (*Cambronero and Puelles, 2000*; *Cramer et al., 2000*; *Marín et al., 2008*; *Marín and Puelles, 1995*). Mouse mapping data from (*Altieri et al., 2015*; *Chen et al., 2012*; *Di Bonito et al., 2015*; *Di Bonito et al., 2013*; *Di Bonito and Studer, 2017*; *Di Bonito et al., 2017*; *Farago et al., 2006*; *Fujiyama et al., 2009*; *Maricich et al., 2009*; *Marrs et al., 2013*; *Marrs and Spirou, 2012*; *Pasqualetti et al., 2007*; *Rose et al., 2009*; *Wang et al., 2005*; *Yamada et al., 2007*). An r3 contribution to nuclei of the superior olivary complex cannot at present be completely ruled out.
DOI: https://doi.org/10.7554/eLife.40232.009

## The analogous avian and mammalian ITD circuits have separate evolutionary and developmental origins

Both mammals and diapsids have brainstem interaural time difference (ITD) sound localisation circuits, with overall similar organisation. However, functional and morphological studies have shown that the similarities are superficial and indicate that the ITD circuits are an example of convergent evolution (*Carr and Christensen-Dalsgaard, 2016*; *Carr and Soares, 2002*; *Grothe et al., 2004*;

*Grothe and Pecka, 2014*; *Grothe et al., 2010*). Our developmental analysis supports this hypothesis by 1) corroborating differences in origin of first order nuclei, 2) revealing divergence in the origins of second order coincidence detectors and 3) showing a multifaceted developmental and evolutionary history for the neurons that provide inhibitory modulation to the ITD circuit.

NM neurons are the bilaterally projecting neurons of the diapsid ITD circuit and they are broadly considered the functional counterpart of the first order spherical bushy cells (SBCs) of the mammalian VCN (*Carr and Soares, 2002*). Both neuronal types have bifurcating ipsilateral and contralateral projections to coincidence detection nuclei. However, SBCs and NM neurons follow different projection trajectories and, while contralateral NM axons are arranged in a delay line that spans the medio-lateral extent of the NL (*Figure 1D,E*) in accordance with the Jeffres model (*Ashida and Carr, 2011*), no such arrangement is present for SBCs axons reaching the contralateral medial superior olive (MSO) (*Karino et al., 2011*). Also, while NM neurons contact exclusively the NL within the auditory brainstem (*Carr and Soares, 2002*; *Takahashi and Konishi, 1988*; *Wild et al., 2010*), SBCs send collateral projections to the lateral superior olive (LSO) and nucleus of the trapezoid body (NTB), in addition to the MSO (*Cant and Benson, 2003*; *Smith et al., 1993*), indicating that they participate in multiple instances of auditory information processing. Finally, while both NM neurons and SBCs arise from $Atoh1^+$ rhombic lip progenitors (*Figures 6A–E,K* and *8* and (*Di Bonito and Studer, 2017*; *Fujiyama et al., 2009*)), their axial origin differs. NM cells originate from the caudal hindbrain, posterior to rhombomere 5 (*Figures 6F–K* and *8* and (*Cambronero and Puelles, 2000*; *Cramer et al., 2000*; *Marín and Puelles, 1995*)), while SBCs originate from rostral rhombomeres (*Di Bonito and Studer, 2017*; *Farago et al., 2006*). This is in agreement with the proposition that NM cells derive from ancestral vestibuloacoustic cells of the caudal octaval column (previous section and (*Carr and Christensen-Dalsgaard, 2016*; *Walton et al., 2017*)), while bilaterally projecting SBCs may be an elaboration of components of an interaural level difference (ILD) circuit, which, in mammals, is proposed to predate the ITD circuit (*Grothe and Pecka, 2014*). The comparison of the developmental origin of NM neurons and SBCs thus supports an independent evolutionary origin for these functionally analogous cell types.

Second order coincidence detection nuclei of mammals and birds (MSO and NL, respectively) show remarkably similar morphologies, with a linear arrangement of bipolar neurons that receive segregated binaural input. (*Figure 1D,E* and (*Carr and Soares, 2002*)). However, the NL and MSO differ in their location within the brainstem (*Figure 1C,D* and (*Grothe et al., 2004*)), employ differing neural coding strategies (*Grothe and Pecka, 2014*; *Grothe et al., 2010*) and most likely have separate evolutionary origins (*Carr and Christensen-Dalsgaard, 2016*; *Grothe et al., 2004*; *Grothe and Pecka, 2014*; *Walton et al., 2017*). In mammals, the MSO is proposed to be an elaboration of the LSO which computes ILDs (*Grothe and Pecka, 2014*). The MSO contains coincidence detection glutamatergic bipolar neurons that derive from $Atoh1^+$ rhombic lip progenitors in r5, in addition to GABAergic neurons of unknown origin (*Altieri et al., 2015*; *Di Bonito and Studer, 2017*; *Maricich et al., 2009*; *Marrs et al., 2013*; *Marrs and Spirou, 2012*; *Rose et al., 2009*). By contrast, we showed here that the avian NL, composed of a single type of bitufted neurons, is derived from $Atoh1^+$ progenitors that overlap with the rostral end of the rhombic lip derived NM at r5 and r6, with an additional contribution from r4 (*Figure 7A–E*). The development of NL neurons progresses within the auditory anlage together with that of NM neurons (*Book and Morest, 1990*; *Hendricks et al., 2006*). On the whole, this supports a common developmental and evolutionary origin for NM and NL neurons of the avian ITD circuit, with both cell types deriving from the ancestral descending octaval nucleus (*Grothe et al., 2004*), and contrasts with the separate developmental and evolutionary histories of first order SBCs and bipolar MSO neurons of the mammalian ITD circuit (*Grothe and Pecka, 2014*).

Finally, inhibitory modulation of coincidence detection neurons is fundamentally different between amniote ITD circuits. In mammals, inhibitory input to MSO neurons is glycinergic, bilateral, phased-locked and provided by LNTB and MNTB neurons (*Grothe, 2003*; *Grothe and Pecka, 2014*; *Grothe et al., 2010*; *Myoga et al., 2014*). In birds, inhibition of NL neurons is GABAergic, unilateral, tonic and provided by the SON (*Burger et al., 2011*; *Grothe, 2003*; *Grothe and Pecka, 2014*; *Grothe et al., 2010*). No individual mammalian nucleus has been identified as equivalent to the diapsid SON (*Burger et al., 2005*; *Burger et al., 2011*; *Grothe et al., 2004*; *Grothe et al., 2010*; *Tabor et al., 2012*; *Wild et al., 2010*). Neurons in the SON originate from both the alar and basal plates, dorsal and ventral parts of the neuroepithelium respectively (*Tan and Le Douarin, 1991*) in

rhombomeres 4 and 5 (*Figures 7E–K* and *8* – (*Cambronero and Puelles, 2000*; *Cramer et al., 2000*; *Marín and Puelles, 1995*), and possibly rhombomere 6, and include a large complement of Atoh1$^+$ rhombic lip derived neurons that, most likely, do not provide the inhibitory modulation to the ITD circuit. Although we could not identify the origin of the GABAergic neurons that comprise about 70% of the SON cells (*Lachica et al., 1994*) and provide descending input to the NL, NM and NA (*Burger et al., 2005*; *Wild et al., 2010*), it is likely that these are specifically born in r5 (*Figures 7* and *8*) and of basal origin (*Tan and Le Douarin, 1991*). Interestingly, this raises the possibility of a common developmental origin with neurons of the mammalian NTB that also originate in rhombomere 5, from basal plate progenitors belonging to the *En-1* lineage (*Altieri et al., 2015*; *Di Bonito and Studer, 2017*; *Maricich et al., 2009*; *Marrs et al., 2013*). Moreover, the mixed developmental origin of the SON resembles that of the mammalian olivary complex as a whole (to which the LSO, MSO and NTB belong) and that is also derived from both basal and alar (including rhombic lip) progenitors. This suggests the possibility of an alar/basal plate derived ancestral amniote second order vestibuloacoustic nucleus located in the ventral hindbrain and composed of multiple neuronal types that would have integrated ascending pathways and provided descending modulatory input to other brainstem vestibuloacoustic nuclei. The extant diapsid SON and mammalian superior olivary complex would thus be independent elaborations of this ancestral second order ventral nucleus.

In summary, the comparative development of ITD circuit components shows a mixture of ancestral and derived features. Subcircuits belonging to the ascending pathways seem to be conserved (e.g. NA and VCN neurons). However, while the mammalian bilateral input (SBC) and coincidence detection neurons (MSO) likely emerged as an elaboration of the ancestral mammalian ILD circuit (*Grothe and Pecka, 2014*), our developmental analysis supports the hypothesis that the avian NM/NL subcircuit may have emerged as an elaboration of a more ancient vertebrate vestibular network (*Carr and Christensen-Dalsgaard, 2016*; *Walton et al., 2017*).

## Materials and methods

### Key resources table

| Reagent type (species) or resource | Designation | Source or reference | Identifiers | Additional information |
|---|---|---|---|---|
| Recombinant DNA reagent | CAG-mCherry | | | Dr.Murakami (Osaka University) |
| Recombinant DNA reagent | Atoh1-Cre | DOI: 10.1523/ JNEUROSCI. 4231–11.2012 | | Avihu Klar, Hebrew University Medical School |
| Recombinant DNA reagent | Atoh1-Gal4 | this paper | | Atoh1 enhancer from Atoh1-Cre subcloned upstream of Gal4 coding sequence (Martin Meyer, King's College London) - Primers in *Table 1* |
| Recombinant DNA reagent | Atoh1-FLPo | this paper | | Atoh1 enhancer from Atoh1-Cre subcloned upstream of FLPo recombinase coding sequence (Avihu Klar, Hebrew University Medical School) - Primers in *Table 1* |

*Continued on next page*

*Continued*

| Reagent type (species) or resource | Designation | Source or reference | Identifiers | Additional information |
|---|---|---|---|---|
| Recombinant DNA reagent | Ptf1a-Cre | this paper | | Ptf1a short enhancer (*Meredith et al., 2009*) - DOI: 10.1523/JNEUROSCI.2303–09.2009) was subcloned upstream of the Cre recombinase coding sequence replacing the Atoh1 enhancer in the Atoh1-Cre plasmid (Avihu Klar, Hebrew University Medical School) - Primers in *Table 1* |
| Recombinant DNA reagent | Egr2-Cre | this paper | | Egr2 enhancer sequence (mm10 chr:67320405–67321006) described in *Chomette et al., 2006* (doi:10.1242/dev.02289) was subcloned upstream of the Cre recombinase coding sequence replacing the Atoh1 enhancer in the Atoh1-Cre plasmid (Avihu Klar, Hebrew University Medical School) - Primers in *Table 1* |
| Recombinant DNA reagent | Hoxb1-Cre | this paper | | Hoxb1 enhancer sequence (mm10 chr11:96365175–96365784) described in *Ferretti et al., 2005* (doi:10.1128/MCB.25.19.8541–8552.2005) was subcloned upstream of the Cre recombinase coding sequence replacing the Atoh1 enhancer in the Atoh1-Cre plasmid (Avihu Klar, Hebrew University Medical School) - Primers in *Table 1* |
| Recombinant DNA reagent | Hoxa3-Cre | this paper | | Hoxa3 enhancer sequence (mm10 chr6:52177190–52177795) described in *Manzanares et al., 2001* (PMID: 9895323) was subcloned upstream of the Cre recombinase coding sequence replacing the Atoh1 enhancer in the Atoh1-Cre plasmid (Avihu Klar, Hebrew University Medical School) - Primers in *Table 1* |

*Continued on next page*

*Continued*

| Reagent type (species) or resource | Designation | Source or reference | Identifiers | Additional information |
|---|---|---|---|---|
| Recombinant DNA reagent | Hoxd4-Cre | this paper | | Hoxd4 enhancer sequence (mm10 chr2:74729772–74731362) described in *Morrison et al. (1997)* (PMID: 9272954) was subcloned upstream of the Cre recombinase coding sequence replacing the Atoh1 enhancer in the Atoh1-Cre plasmid (Avihu Klar, Hebrew University Medical School) - Primers in *Table 1* |
| Recombinant DNA reagent | Pbase | Sanger Institute | | |
| Recombinant DNA reagent | pCAG-LoxP-pA-LoxP-EGFP | DOI: 10.1523/ JNEUROSCI.4231–11.2012 | | Avihu Klar, Hebrew University Medical School |
| Recombinant DNA reagent | pCAG-FRT-pA-FRT-LoxP-pA-LoxP-EGFP | DOI: 10.1093/nar/gku750 | | Avihu Klar, Hebrew University Medical School |
| Recombinant DNA reagent | UAS-tdT | | | Martin Meyer (King's College London) |
| Recombinant DNA reagent | chick *Atoh1* riboprobe | | | *Wilson and Wingate, 2006* (doi:10.1016/ j.ydbio.2006.05.028) |
| Recombinant DNA reagent | chick *Ptf1a* riboprobe | | ChEST1028o4 | *Green and Wingate, 2014* (doi:10.1242/ dev.099119) |
| Antibody | rabbit polyclonal anti-GFP | Invitrogen -ThermoFisher | Cat no. A11122 | IHC (1:500) |
| Antibody | Alexa 488-conjugated goat polyclonal anti-rabbit IgG | Molecular Probes - ThermoFisher | Cat no. A11034 | IHC (1:500) |
| Commercial assay or kit | Gibson Assembly kit | New England Biolabs | Cat no. E5510S | |

## Cloning of enhancer elements and plasmids

An *Atoh1* enhancer (*Helms et al., 2000*) was used to direct Cre recombinase expression to cells derived from the rhombic lip (*Kohl et al., 2012*). The same element was also subcloned upstream the Gal4 coding sequence (kind gift from Martin Meyer, King's College London) or the FLPo recombinase coding sequence (*Hadas et al., 2014*) to allow for the intersectional labelling of cells. A *Ptf1a* short enhancer element (*Meredith et al., 2009*) was subcloned upstream the Cre recombinase cloning sequence and used to identify a population of cells originating from the ventricular zone. All subclonings were performed using the Gibson Assembly kit (New England Biolabs).

For the labelling of cells originating at different antero-posterior levels along the hindbrain, a set of rhombomere specific enhancer elements were employed. An *Egr2* (*Krox20*) enhancer element (*Chomette et al., 2006*) was used to direct Cre recombinase expression to r3 and r5. An enhancer element from *Hoxb1* (*Ferretti et al., 2005*) was used to direct Cre expression to r4. A *Hoxa3* enhancer element (*Manzanares et al., 2001*) was used to direct Cre expression to r5/r6. Finally, an enhancer element from *Hoxd4* (*Morrison et al., 1997*) was used to direct Cre expression to >r7. The enhancer elements were PCR amplified from mouse genomic DNA, and subcloned upstream of

**Table 1.** Restriction enzymes and sequences of the oligos used for cloning of expression constructs.
Uppercase, gene-specific portion. Lowercase, vector specific portion.

| Plasmid | PCR/digest | Restriction enzymes or primers sequences |
|---|---|---|
| Atoh1-Gal4 | Vector | NcoI + NotI digest of pBαtubGal4 vector |
| | Insert PCR | ctccaccgcggtggcAGAGCTTCCACTTCACCTCTCTGAGTG |
| | On Atoh1-Cre vector | gtttcttcttgggcccGGGGAGCGGCGAGAGGCT |
| Atoh1-FLPo | Vector | NcoI + NotI digest of CAG-FLPo vector |
| | Insert PCR | agcagagcgcggcgcCTCCTGGGCAACGTGCTG |
| | On Atoh1-Cre vector | cctgaggagtgaattggcGAATTCCTCATCAGATCCGCC |
| Ptf1a-Cre | Vector | NcoI + SacI digest of Atoh1-Cre vector |
| | Insert PCR | gggcgaattggagctAGGATCGTCAGCCACAGAGTTCATGG |
| | On Ptf1a-GFP vector | ctgcagatatccagccCATGGCGCCGCGCTCTGC |
| Egr2-Cre | Vector | SacI + XmaI digest of Atoh1-Cre vector |
| | Insert PCR | gggcgaattggagctGGGTTGTGAATGGAGCCAG |
| | On mouse gDNA | attcctgcagcccggGCAAGCCGACCAAACTCC |
| Hoxb1-Cre | Vector | SacI + XmaI digest of Atoh1-Cre vector |
| | Insert PCR | gggcgaattggagctCTAGTCATCCTTTTGTCCC |
| | On mouse gDNA | attcctgcagcccggTCTTGCCCTACAACCTTTC |
| Hoxa3-Cre | Vector | SacI + XmaI digest of Atoh1-Cre vector |
| | Insert PCR | gggcgaattggagctATCAAATAGCAGCGAATCTTCG |
| | On mouse gDNA | attcctgcagcccggGGGACGTGTAGGAGGTGA |
| Hoxd4-Cre | Vector | SacI + XmaI digest of Atoh1-Cre vector |
| | Insert PCR | gggcgaattggagctCTAGAAGCCCACAGAAGTTG |
| | On mouse gDNA | attcctgcagcccggCTAGAGCAGGTTCCCAGATG |

DOI: https://doi.org/10.7554/eLife.40232.010

the Cre recombinase coding sequence using the Gibson Assembly kit (New England Biolabs). Primers used for each step are listed in *Table 1*.

The different enhancer driven expression plasmids were co-electroporated with their corresponding reporter plasmids, driving Cre, FLPo or Gal4 dependent expression of fluorescent proteins. The following reporter plasmids were employed: pCAG-LoxP-pA-LoxP-EGFP (CAG-Flox-pA-GFP), pCAG-FRT-pA-FRT-LoxP-pA-LoxP-EGFP (CAG-Flox-FLp-pA-GFP) and UAS-tdT. On experiments using *Ptf1a*, *Egr2*, *Hoxb1*, *Hoxa3* and *Hoxd4* enhancers, the PiggyBac DNA-transposition system was used to achieve integration of the reporter gene into the chick genome and therefore persistent labelling of the targeted cells (*Hadas et al., 2014*). The reporter construct in this case consisted of a Cre-dependent GFP cloned between the two PB arms and was co-electroporated with an enhancer-Cre plasmid and a plasmid encoding for the Pbase transposase (Sanger Institute). As a control for the successfully targeted area, a CAG-mCherry plasmid was co-electroporated in most experiments. All plasmids and sequences are available upon request.

### *In ovo* electroporations

Fertilised hen's eggs were incubated at 38°C. Electroporations were performed at stages HH12-15 (*Hamburger and Hamilton, 1951*). Briefly, eggs were windowed using sharp surgical scissors. The fourth ventricle was injected with ~100–200 nl of the corresponding plasmids DNA at equimolar concentrations and to a final concentration of 1–3 µg/µl. Three 20 ms/10 V square waveform electrical pulses were passed between electrodes placed on either side of the hindbrain. Tyrode's solution supplemented with penicillin/streptomycin (Sigma) was added before the eggs were resealed and incubated for a further 1 or 8 days at 38°C. A minimum of three batches of independent electroporations were performed for each of the enhancer plasmid combinations employed. A minimum of two embryos were fully processed (dissected, sectioned, stained, imaged and analysed) for each of the

**Table 2.** Summary of the plasmid combinations used, the structures labelled, the figures showing representative images of the labelling observed and the number of embryos analysed.

| Enhancer/reporter plasmid combinations | Abbreviation | Structures labelled (Figures) – [# embryos] |
|---|---|---|
| Hindbrain flatmounts (embryos fixed at E4/E6) | | |
| Atoh1-Cre + CAG Flox-pA-GFP+CAG-mCherry | Atoh1::GFP + mCherry | RL (2C) – (12) |
| Ptf1a-Cre + CAG PBase+Pb CAG-Flox-pA-GFP+CAG-mCherry | Ptf1a::GFP + mCherry | VZ (2D) – (7) |
| Egr2-Cre + CAG PBase+Pb CAG-Flox-pA-GFP+CAG-mCherry | Egr2::GFP + mCherry | r3, r5 (2E) – (5) |
| Hoxb1-Cre + CAG PBase+Pb CAG-Flox-pA-GFP+CAG-mCherry | Hoxb1::GFP + mCherry | r4 (2E) – (5) |
| Hoxa3-Cre + CAG PBase+Pb CAG-Flox-pA-GFP+CAG-mCherry | Hoxa3::GFP + mCherry | r5, r6 (2E) – (5) |
| Hoxd4-Cre + CAG PBase+Pb CAG-Flox-pA-GFP+CAG-mCherry | Hoxd4::GFP + mCherry | ≥r7 (2E) – (5) |
| Hindbrain coronal cryosections (embryos fixed at E10) | | |
| Atoh1-Cre + CAG Flox-pA-GFP+CAG-mCherry | Atoh1::GFP + mCherry | VeS (4A), Dd/Dv (4E-F), Ta (4K), VeM (5A), VeD (5H-I), NM (6A-D), NA (6 L-M), NL (7A), SON (7F) – (10) |
| Atoh1-Cre + CAG-Flox-pA-GFP | Atoh1::GFP | NL (7A - inset), SON (7F - inset) – (4) |
| Ptf1a-Cre + CAG PBase+Pb CAG-Flox-pA-GFP+CAG-mCherry | Ptf1a::GFP + mCherry | Dd/Dv (4 G-6H), Ta (4L), VeM (5B), VeD (5J), -NM (6E), NA (6N), SON (7G) – (9) |
| Egr2-Cre + CAG PBase+Pb-CAG-Flox-pA-GFP | Egr2::GFP | VeS (4B), Dd/Dv (4I), Ta (4M), VeD (5K), NA (6O), NL (7B), SON (7H) – (4) |
| Hoxb1-Cre + CAG PBase+Pb-CAG-Flox-pA-GFP | Hoxb1::GFP | VeS (4C), Dd/Dv (4J), Ta (4N), VeD (5N), NA (6P), NL (7C), SON (7J) – (7) |
| Hoxa3-Cre + CAG PBase+Pb-CAG-Flox-pA-GFP | Hoxa3::GFP | Ta (4O), VeD (5L), NL (7D), SON (7K) – (4) |
| Hoxd4-Cre + CAG PBase+Pb-CAG-Flox-pA-GFP | Hoxd4::GFP | VeD (5M) – (7) |
| Egr2-Cre + CAG PBase+Pb CAG-Flox-pA-GFP+Atoh1 Gal4+UAS-tdT | Egr2::GFP + Atoh1::tdT | VeM (5C), NM (6F), SON (7I)– (4) |
| Hoxb1-Cre + CAG PBase+Pb CAG-Flox-pA-GFP+Atoh1 Gal4+UAS-tdT | Hoxb1::GFP + Atoh1::tdT | VeM (5D), -NM (6H)– (7) |
| Hoxa3-Cre + CAG PBase+Pb CAG-Flox-pA-GFP +Atoh1 Gal4+UAS-tdT | Hoxa3::GFP + Atoh1::tdT | VeM (5E), NM (6G) – (4) |
| Hoxd4-Cre + CAG PBase+Pb CAG-Flox-pA-GFP+Atoh1 Gal4+UAS-tdT | Hoxd4::GFP + Atoh1::tdT | VeM (5F), NM (6I) – (7) |
| Hoxd4-Cre + Atoh1 FLPo+CAG Flox-FLp-pA-GFP+CAG-mCherry | Hoxd4 + Atoh1::GFP + mCherry | NM (6J) – (2) |

DOI: https://doi.org/10.7554/eLife.40232.011

electroporations performed. *Table 2* shows a summary of the plasmid combinations used, the structures labelled, the figures showing representative images of the labelling observed and the number of embryos analysed.

## Tissue processing, in situ hybridisation, immunostaining and imaging

Embryos were collected after 3 to 10 days of incubation, the hindbrain was dissected out and fixed in 4% paraformaldehyde (in phosphate-buffered saline). For *in situ* hybridisation E4 dissected hindbrain tissue was stained as previously described (*Myat et al., 1996*) with digoxygenin or fluorescein-labelled riboprobes (Roche) for: *Atoh1* (*Wilson and Wingate, 2006*) and *Ptf1a* (ChEST1028o4, (*Green et al., 2014*)), flatmounted on 80% glycerol and imaged from the dorsal side.. A different set of E10 dissected hindbrains were sectioned at 10 um and stained with cresyl violet.

Electroporated E3/E6 wholemount hindbrains were flatmounted on 80% glycerol and imaged from the dorsal side. Electroporated E10 hindbrains were cryoprotected using a sucrose gradient, embedded in OCT (VWR) and frozen on liquid nitrogen. Floating coronal cryosections of 80 μm thickness were washed three times in PBS. All sections were counterstained with NucRed (Molecular Probes). Sections labelled for *Egr2*, *Hoxb1*, *Hoxa3* or *Hoxd4* were immunostained for GFP. Briefly, floating cryosections were washed three times 30 min in PBS/1% TritonX-100 (PBSTx) before being

washed in a blocking solution of 10% goat serum in PBSTx twice for one hour at room temperature. Anti-GFP antibody (rabbit IgG - 1:500, Invitrogen) was diluted in blocking solution and the sections were incubated at 4°C for 3 days. Samples were then rinsed in blocking solution and washed three times for one hour in blocking solution before adding an Alexa 488-conjugated goat anti-rabbit IgG, at 1:500 (Molecular Probes) diluted in blocking solution and incubating at 4°C for a further 2 days. Sections were then washed three times in PBS and nuclei were stained with NucRed (Molecular Probes). Sections were mounted on Prolong Diamond (Molecular probes).

Digital brightfield images were acquired on a stereo microscope (Leica MZFLIII). Laser scanning confocal microscopy images were acquired on an Olympus AX70 microscope. Image analysis and processing was performed in ImageJ and Photoshop.

All reports of positive labelling of cells belonging to a specific lineage, labelled by a given enhancer, at a given nucleus represent at least three independent observations in embryos obtained from three independent electroporations.

## Acknowledgements

We would like to thank Dr Martin Meyer and Dr Avihu Klar for kindly providing plasmid constructs and Prof Anthony Graham for invaluable comments on the manuscript. This work was supported by a Newton International Fellowship (Royal Society) to ML.

## Additional information

### Funding

| Funder | Grant reference number | Author |
| --- | --- | --- |
| Royal Society | NF120319 | Marcela Lipovsek |

The funders had no role in study design, data collection and interpretation, or the decision to submit the work for publication.

### Author contributions

Marcela Lipovsek, Conceptualization, Resources, Formal analysis, Funding acquisition, Investigation, Visualization, Methodology, Writing—original draft, Project administration, Writing—review and editing; Richard JT Wingate, Conceptualization, Resources, Project administration, Writing—review and editing

### Author ORCIDs

Marcela Lipovsek (iD) http://orcid.org/0000-0001-9328-0328
Richard JT Wingate (iD) https://orcid.org/0000-0002-1662-6097

### Decision letter and Author response

Decision letter https://doi.org/10.7554/eLife.40232.014
Author response https://doi.org/10.7554/eLife.40232.015

## Additional files

### Supplementary files

• Transparent reporting form
DOI: https://doi.org/10.7554/eLife.40232.012

### Data availability

Images included in Figures 2-7 are representative of all the data generated and analysed during this study.

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
