## [Decision Letter]

Thank you for submitting your article "Conserved and divergent development of brainstem vestibuloacoustic nuclei" for consideration by *eLife*. Your article has been reviewed by three peer reviewers, including Catherine Emily Carr as the Reviewing Editor and Reviewer #1, and the evaluation has been overseen by Andrew King as the Senior Editor. The following individuals involved in review of your submission have agreed to reveal their identity: Karina Cramer (Reviewer #2); Hans-Gerd Nothwang (Reviewer #3).

The reviewers have discussed the reviews with one another and the Reviewing Editor has drafted this decision to help you prepare a revised submission.

Summary:

We assume that when vertebrates moved onto land, they were insensitive to airborne sound. Paleontological and evodevo studies have indicated that the structures associated with sensitive hearing (middle ear bones, a tympanum) evolved only after the major tetrapod lineages diverged, i.e. tympanic hearing evolved in parallel. It is further assumed that invention of tympana would have led to elaboration of the central auditory nuclei. Just how these changes occurred continues to be the subject of speculation. This paper represents a new approach to unraveling the contribution of brainstem lineages to the development of the chick hindbrain vestibular and auditory nuclei. The authors used the vestibular system as a control – it's fairly stable over evolution – and compared the chick developmental map with genetic fate-maps of the mouse hindbrain. They found similar origins for vestibular nuclei in chick and mouse, while the auditory nuclei show a mixture of conserved and divergent origins.

Essential revisions:

Detailed, useful comments are found in the reviews, appended below. Most are fairly minor suggestions to improve the clarity of the text or improve the analysis. One reviewer suggested you consider changing the order of the result sections by starting with the vestibular system.

Title: You might wish to make your title more broadly accessible.

Reviewer #2:

This manuscript presents a detailed and very elegant study using genetic fate mapping to identify the origins of vestibular and auditory nuclei in the hindbrain of the chick embryo and presents a comparison with genetic fate maps obtained in mouse. It is especially significant in that most of the previous studies used dye labeling or grafts, whereas the mouse studies are based on genetic labeling. The study thus allows for a meaningful comparison taking into account rhombomeric origin as well as two key transcription factors, *Ptf1a* and *Atoh1*, which are expressed in the ventricular zone and the rhombic lip, respectively. The data support the overall conclusion that vestibular lineages are conserved while the auditory lineages are divergent.

Reviewer #3:

Longtime, neuronal populations in the tetrapod auditory hindbrain were considered to represent homologues, due to a conserved bauplan of the system. Yet, the seminal discovery that the tympanic inner ear likely evolved several times independently raised serious questions concerning this assumption. Morphological and functional analyses supported the conjecture of independent, convergent evolution. Yet, the issue is far from settled and modern genetic approaches hold great promise to contribute significantly to this ongoing debate. The manuscript of Lipvsek and Wingate present a powerful example of such contributions. Using state of the art enhancer constructs together with electroporation, they performed lineage tracing the vestibular and auditory nuclei in the chicken hindbrain. The data, in combination with the extensive literature on the murine cell fate, allowed the authors to provide intriguing insights into evolutionary trajectories and to sketch a new scenario for the evolution of the tetrapod vestibuloacoustic nuclei in the hindbrain. Most of the conclusions are well supported by the data and well presented. Overall, the manuscript significantly advances our knowledge on the development and evolution of the vestibular and auditory hindbrain.

I have only few concerns and suggestions.

The SON is strongly labeled by Atoh1::GFP (Figure 5F and text). This is surprising in view of the fact that these cells should constitute less than 30% of the SON. The authors should present a cell count (e.g. Atoh1+ versus CAG::mCherry+ cells) to clarify this issue.

A major suggestion of the paper is that the avian NA and mammalian VCN are homologous structures. This would assume that SBCs, which are major output neurons of the VCN, are homologous to cells of the NA. On one hand, this fits the authors' data that SBC and avian NM neurons are derived from different rhombomeres and hence are not homologous. On the other hand, to my knowledge, a spherical bushy cell type has not been described in the NA. Could the authors comment on this, and how this fits into their general scheme of the evolution of first order auditory nuclei in tetrapods? For previous reviews on this issue see Grothe et al., 2004, Nothwang, 2016.

Furthermore, the author conclude that SBCs may be an elaboration of components of an interaural level difference circuit, which is proposed to predate in mammals the ITD circuit. Note, however, that both ILD (LSO) and ITD (MSO) processing cells of the circuits are derived from r5 and not from r2-4, as do SBCs. The authors should therefore define more explicitly, what they mean by "elaboration" and consider rephrasing the sentence.

[Editors' note: further revisions were requested prior to acceptance, as described below.]

Thank you for resubmitting your article "Conserved and divergent development of brainstem vestibular and auditory nuclei" for consideration by *eLife*. Your revised article has been reviewed by three peer reviewers, including Catherine Emily Carr as the Reviewing Editor and Reviewer #1, and the evaluation has been overseen by Andrew King as the Senior Editor. The following individuals involved in review of your submission have agreed to reveal their identity: Karina Cramer (Reviewer #2); Hans-Gerd Nothwang (Reviewer #3).

The reviewers have discussed the reviews with one another and the Reviewing Editor has drafted this decision to help you prepare a revised submission. We thought you did an excellent job in the revisions. We only have a few remaining points that you might like to address.

Figure 1C shows the auditory nerve entering the inside of the U shape. This is the efferent tract, nerve inputs are dorsal, so in the interest of accuracy, please modify the diagram.

Citations: You might want to include citations to one or more of the Peusner papers.

Popratiloff and Peusner (2007). Otolith fibers and terminals in chick vestibular nuclei. J Comp Neurol 502:19-37.

Peusner and Morest (1977). A morphological study of neurogenesis in the nucleus vestibularis tangentialis of the chick embryo. Neurosci 2:209-227.

With respect to your reply to reviewer 3, you write "Also to our knowledge, no SBC-type of neuron has been described in the NA that shares all these criteria." The Soares papers describe a stubby cell that matches most SBC criteria.

---

## [Author Response]

Essential revisions:Detailed, useful comments are found in the reviews, appended below. Most are fairly minor suggestions to improve the clarity of the text or improve the analysis. One reviewer suggested you consider changing the order of the result sections by starting with the vestibular system.

This is an excellent suggestion and we have changed the order of the results accordingly. This change makes indeed for clearer reading. Note: for clarity, the re-ordering of sections has not been included in the track changes. Figures have been re-numbered accordingly throughout the text and this is recorded in the track changes.

Title: You might wish to make your title more broadly accessible.

We have given this issue a great deal of thought and decided to change the Title to “Conserved and divergent development of brainstem vestibular and auditory nuclei”.

Reviewer #3:Longtime, neuronal populations in the tetrapod auditory hindbrain were considered to represent homologues, due to a conserved bauplan of the system. Yet, the seminal discovery that the tympanic inner ear likely evolved several times independently raised serious questions concerning this assumption. Morphological and functional analyses supported the conjecture of independent, convergent evolution. Yet, the issue is far from settled and modern genetic approaches hold great promise to contribute significantly to this ongoing debate. The manuscript of Lipvsek and Wingate present a powerful example of such contributions. Using state of the art enhancer constructs together with electroporation, they performed lineage tracing the vestibular and auditory nuclei in the chicken hindbrain. The data, in combination with the extensive literature on the murine cell fate, allowed the authors to provide intriguing insights into evolutionary trajectories and to sketch a new scenario for the evolution of the tetrapod vestibuloacoustic nuclei in the hindbrain. Most of the conclusions are well supported by the data and well presented. Overall, the manuscript significantly advances our knowledge on the development and evolution of the vestibular and auditory hindbrain.I have only few concerns and suggestions.The SON is strongly labeled by Atoh1::GFP (Figure 5F and text). This is surprising in view of the fact that these cells should constitute less than 30% of the SON. The authors should present a cell count (e.g. Atoh1+ versus CAG::mCherry+ cells) to clarify this issue.

We acknowledge that the labelling of Atoh1 neurons in the SON is strong and surprising in the context of published literature that shows that ~70% of the neurons in this nucleus are GABAergic. We have performed pilot immunostainings on Atoh1 labelled samples and our preliminary data shows that the Atoh1+ neurons in the SON are not GABAergic and so should comprise less than 30% of the cells in the nucleus.

The Atoh1 labelling does appear to cover most of the nucleus in Figure 5F. However, this results from the Atoh1+ neurons having many projections within the nucleus. For example, the z-stack projection in Figure 5F has 33 Atoh1+ neurons in total, and the inset figure has only 15 (10.5% and 4.6% of total cells, respectively). Although we have avoided this type of quantification given the technical variabilities inherent to the electroporation labelling technique (see above), we have now counted the number of Atoh1+ cells in successfully labelled SONs from 9 coronal slices from 4 independent electroporations (as a percentage of the total number of cells in the nucleus assessed by nuclear staining). These informal numbers show that the percentage of Atoh1+ neurons in the SON is 9.5 ± 2.1% (mean ± SEM), with 13.8% the highest value obtained. In summary, and as stated in the Discussion section, Atoh1+ neurons in the SON constitute a new and intriguing population that may be potentially involved in the binaural integration of auditory information and also provide descending modulatory input to other brainstem vestibuloacoustic nuclei.

A major suggestion of the paper is that the avian NA and mammalian VCN are homologous structures. This would assume that SBCs, which are major output neurons of the VCN, are homologous to cells of the NA. On one hand, this fits the authors' data that SBC and avian NM neurons are derived from different rhombomeres and hence are not homologous. On the other hand, to my knowledge, a spherical bushy cell type has not been described in the NA. Could the authors comment on this, and how this fits into their general scheme of the evolution of first order auditory nuclei in tetrapods? For previous reviews on this issue see Grothe et al., 2004, Nothwang, 2016.

We have tried to avoid homologising these two nuclei, although we do stress the general hypothesis that “anterior neurons of the ascending auditory pathwayof amniotesmay be homologous”. To be more specific, dorsal neurons of the anterior octaval column, that receive direct input from the auditory nerve and make ascending projections, may have a common evolutionary origin.

SBCs of the mammalian VCN are characterised by their branching ipsi and contra projections to brainstem nuclei and by their fast and reliable transmission that allows them to remain phase-locked at high stimulus frequencies. Also, to our knowledge, no SBC-type of neuron has been described in the NA that shares all these criteria. This type of cell, within the anterior octaval column, is a novelty of the mammalian VCN. However, the NA does have neurons that project within the brainstem, to the SON (and mostly ipsilaterally), although this has not yet been correlated to a particular cell type within the nucleus. It would be interesting to simultaneously study the morphology, projection pattern and expression of markers such as Atoh7, calretinin and parvalvumin, together with Atoh1 vs Ptf1a labelling. These are all very interesting questions that are beyond the scope of our present work.

Furthermore, the author conclude that SBCs may be an elaboration of components of an interaural level difference circuit, which is proposed to predate in mammals the ITD circuit. Note, however, that both ILD (LSO) and ITD (MSO) processing cells of the circuits are derived from r5 and not from r2-4, as do SBCs. The authors should therefore define more explicitly, what they mean by "elaboration" and consider to rephrasing the sentence.

The hypothesis that the mammalian ITD circuit derives from components of the more ancestral mammalian ILD circuit was put forward by Grothe and collaborators (Grothe, 2010; Grothe, 2014) and is not a conclusion of our paper. This has now been more clearly stated in the Discussion section by rephrasing the sentence: “This is in agreement with the proposition that NM cells derive from ancestral vestibuloacoustic cells of the caudal octaval column (previous section and (Carr and Christensen-Dalsgaard, 2016; Walton et al., 2017)), while bilaterally projecting SBCs may be an elaboration of components of an interaural level difference (ILD) circuit, which, in mammals, is proposed to predate the ITD circuit (Grothe and Pecka, 2014).”

[Editors' note: further revisions were requested prior to acceptance, as described below.]

The reviewers have discussed the reviews with one another and the Reviewing Editor has drafted this decision to help you prepare a revised submission. We thought you did an excellent job in the revisions. We only have a few remaining points that you might like to address.Figure 1C shows the auditory nerve entering the inside of the U shape. This is the efferent tract, nerve inputs are dorsal, so in the interest of accuracy, please modify the diagram.

This was an unintentional error. The diagram in Figure 1C has now been corrected and matches the connectivity outlined in Figure 1E.

Citations: You might want to include citations to one or more of the Peusner papers.Popratiloff and Peusner (2007). Otolith fibers and terminals in chick vestibular nuclei. J Comp Neurol 502:19-37.Peusner and Morest (1977). A morphological study of neurogenesis in the nucleus vestibularis tangentialis of the chick embryo. Neurosci 2:209-227.

Both papers listed describe vestibuloacoustic afferent pathways that we do not investigate in this paper. For this reason, they were not cited in the manuscript.

With respect to your reply to reviewer 3, you write "Also to our knowledge, no SBC-type of neuron has been described in the NA that shares all these criteria." The Soares papers describe a stubby cell that matches most SBC criteria.

The Soares papers describe neurons in the NA that resemble SBC morphological and/or electrophysiological properties. However, the branching ipsilateral and contralateral projections that are the defining feature of SBCs of the ITD circuit in mice have not yet been described for NA cells in chick.